# Self-assembled polyelectrolytes with ion-separation accelerating channels for highly stable Zn-ion batteries

Xueying Hu[1], Haobo Dong [1,2] ✉, Nan Gao[3], Tianlei Wang[1], Hongzhen He[1], Xuan Gao[1], Yuhang Dai [1], Yiyang Liu[4], Dan J. L. Brett[4], Ivan P. Parkin [1] & Guanjie He [1] ✉

Aqueous zinc-ion batteries offer a sustainable alternative to lithium-ion batteries due to their abundance, safety, and eco-friendliness. However, challenges like hydrogen evolution and uncontrolled diffusion of $H^+$, $Zn^{2+}$, and $SO_4^{2-}$ in the electrolyte lead to the dendrite formation, side reactions, and reduced Coulombic efficiency for Zn nucleation. Here, to simultaneously regulate the diffusion of cations and anions in the electrolyte, an ion-separation accelerating channel is constructed by introducing layer-by-layer self-assembly of a flocculant poly(allylamine hydrochloride) and its tautomer poly(acrylic acid). The dual-ion channels, created by strong electrostatic interactions between carboxylate anions and ammonia cations, block $SO_4^{2-}$ and promote the uniform Zn deposition along the Zn(002) plane, exhibiting a CE of 99.8% after 1600 cycles in the Cu∥Zn cell. With the facile fabrication of the layer-by-layer self-assembled Zn anode, an Ah-level pouch cell (17.36 Ah) with a high mass loading (> 8 mg cm$^{-2}$) demonstrates the practical viability for large-scale applications, retaining a capacity of 93.6% for 250 cycles at 1.7 C (35.3 min). This work enables more uniform Zn deposition and enhances the cycling stability in larger pouch cells, paving the way for the commercialisation of zinc-ion batteries.

Aqueous zinc-ion batteries (AZIBs) are regarded as one of the most promising alternatives to lithium-ion batteries for grid-scale electrochemical energy storage (EES) systems due to their high volumetric capacity (5855 mAh cm$^{-3}$), low redox potential (−0.762 V vs standard hydrogen electrode (SHE)), and high safety[1,2]. However, the hydrogen evolution reaction (HER) leads to a rapid rise in the local concentration of $OH^-$ at the anode/electrolyte interface, which further reacts with $SO_4^{2-}$ in the electrolyte to form the by-product ($Zn_4SO_4(OH)_6 \cdot xH_2O$, ZHS)[3]. Moreover, the generation of the inert by-product reduces the active sites for Zn deposition, increases the nucleation barrier, and causes uncontrollable dendrite growth on the Zn anode. This results in a shortened cycle life and has hindered the commercial application of ZIBs[4].

Surface modification using inorganic and organic coatings can effectively inhibit the dendrite growth and side reactions at the Zn anode[5,6]. Inorganic coatings, including carbon-based materials, eutectic alloys, and metallic compounds (e.g., carbon dots (CDs)[7], Zn-Cu[8], Zn-Sn[9], CaCO$_3$[10], ZnF$_2$[11]) can be used as physical barriers to protect the Zn anode from corrosion and regulate $Zn^{2+}$ diffusion to achieve uniform Zn deposition. However, the non-uniform physical barriers lead to a low $Zn^{2+}$ conductivity and a significant volume change during plating/stripping, ultimately causing cracking and peeling[12]. In

[1]Christopher Ingold Laboratory, Department of Chemistry, University College London, London, UK. [2]School of Future Technology, South China University of Technology, Guangzhou, Guangdong, China. [3]State Key Lab of Superhard Materials, College of Physics, Jilin University, Changchun, China. [4]Hanwei Co., Ltd., Building A6, Guoke Artificial Intelligence Innovation Center, Zhejiang, China. ✉e-mail: dhbhubble@scut.edu.cn; g.he@ucl.ac.uk

contrast, the flexible organic polyelectrolyte coatings such as polyamide[13], polyacrylamides[14], and poly(2-vinylpyridine)[15] with 3D cross-linked polymer channels can provide active sites to facilitate $Zn^{2+}$ transference and reduce the interface resistance[16]. But the mono polyelectrolyte interface cannot satisfactorily control the ion diffusion and offer sufficient mechanical strength. For instance, although anionic polyelectrolytes can effectively regulate $Zn^{2+}$ flux to homogenous deposition, it has a limited repelling effect on $SO_4^{2-}$ in the electrolyte, which would cause the by-product formation to a certain extent[17,18]. Owing to the repulsion between anionic polyelectrolytes and the negatively charged Zn anode, the adhesivity of the coatings is also not satisfactory. Based on this, the layer-by-layer (LbL) self-assembly of polyelectrolytes with the controllable composition and tunable properties, which allows sequential deposition of versatile polycations and polyanions on a charged substrate, is an attractive approach to enhance the overall performance of Zn anodes[19–21]. The strong electrostatic interactions between polycations and polyanions can provide oppositely charged dual-ion channels to suppress the corrosion and passivation on the Zn anode, enhancing the mechanical strength (e.g., toughness, adhesion, self-healing)[22–24]. The resources of polycations and polyanions with the characteristics of non-toxic, biocompatible, and biodegradable for the LbL self-assembled SEI layer are highly abundant, which is not only conducive to the preparation of multifunctional SEI layers through the modification of polyelectrolytes but also can promote the development of eco-friendly Zn-ion batteries. The LbL method also allows precise control of the thickness and composition of coatings, making it a sustainable and effective approach compared to other surface modification techniques[25,26]. Moreover, the LbL self-assembly technique applied to prepare PAH/PAA multilayers is more cost-effective for practical application due to its simple manufacturing process and low demand for equipment[27]. It can be readily implemented using roll-to-roll or extrusion-based coating systems, which are already widely used in battery manufacturing[28,29]. A detailed cost breakdown for this LbL self-assembled PAH/PAA multilayer strategy at both lab and projected industrial scales was calculated in Table S6, highlighting the economic feasibility and potential cost reductions with industrial production.

However, limited studies have been conducted on using the LbL self-assembly technique for the interface engineering of Zn anodes. Identifying efficient and appropriate polyelectrolyte combinations for the LbL self-assembled layers remains a challenge, as is demonstrating their effectiveness in protecting Zn anodes and enhancing their applications.

To overcome the above-mentioned challenges, we selected poly(allylamine hydrochloride) (PAH) and its tautomer poly(acrylic acid) (PAA) to prepare the LbL self-assembled PAH/PAA multilayers. Due to the tautomerisation, the photon exchange occurs between the carboxylic acid group (−COOH) of PAA and the amine group (−RNH₂) of PAH, leading to the negatively charged carboxylate group (−COO⁻) and the positively charged ammonia group (−RNH₃⁺) with the strong electrostatic interactions[30]. Based on this, the PAH/PAA multilayers can be seen as the dual-ion channels for $SO_4^{2-}$ and $Zn^{2+}$ in the electrolyte, like ionic separation, the dual-ion channels block $SO_4^{2-}$ and attract $Zn^{2+}$, which regulates the mobility and dispersion of $Zn^{2+}$ and suppresses the side reactions (Fig. 1a). Moreover, the strong electrostatic interactions between PAH and PAA can effectively improve the mechanical strength without affecting the ionic conductivity of the coating and also simplify the preparation process[21,31]. As illustrated in Fig. 1b, the preparation sequence of multilayers is to first coat the PAH layer and then the PAA layer (Anode⁻ − PAH⁺ − PAA⁻), followed by a rinsing process after each coating to remove the weakly associated bound chains. Designing in the sequence: Anode⁻ − PAH⁺ − PAA⁻ would enable a high adhesion to the negatively charged Zn anode and increase the zincophilicity of multilayers. PAA layer as the outer layer can first regulate the diffusion of $Zn^{2+}$ and repel $SO_4^{2-}$ to a certain degree, while PAH layer can further capture $SO_4^{2-}$ due to the low binding energy. Indeed, the PAH/PAA multilayers lead to the formation of ion-separation accelerating channels to block $SO_4^{2-}$ and accelerate $Zn^{2+}$ transference, thereby promoting the uniform Zn deposition and inhibiting the HER and by-products. Remarkably, the LbL self-assembled PAH/PAA multilayers are favourable for the preferential nucleation and growth of $Zn^{2+}$ along the Zn(002) surface to form a smooth and dense deposition layer and suppress the dendrite formation (Fig. 1c). Correspondingly, the PAH/PAA multilayers enable a

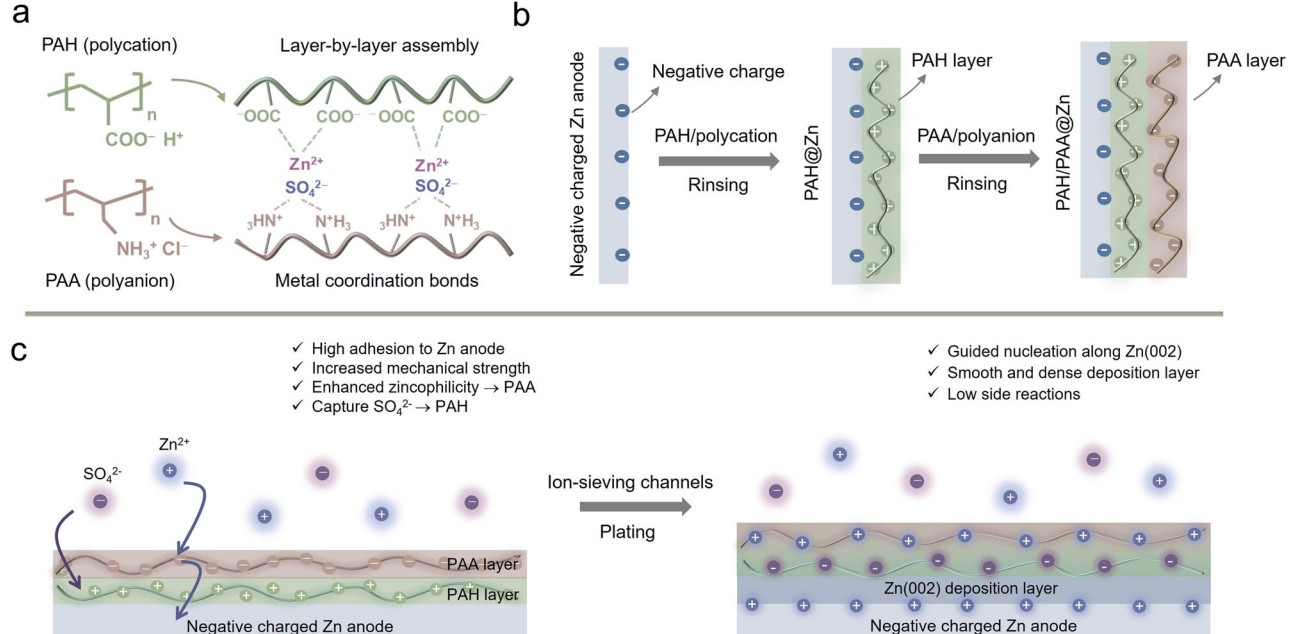

Fig. 1 | Schematic diagram of the ion-separation Zn anode by layer-by-layer method. a The $Zn^{2+}/SO_4^{2-}$ ion-sieving accelerating channel model generated by the PAH/PAA electrostatic interaction. b The preparation process of the LbL self-assembly PAH/PAA multilayers on Zn anodes. c Diffusion and deposition behaviours of $Zn^{2+}$ on the PAH/PAA coating surface.

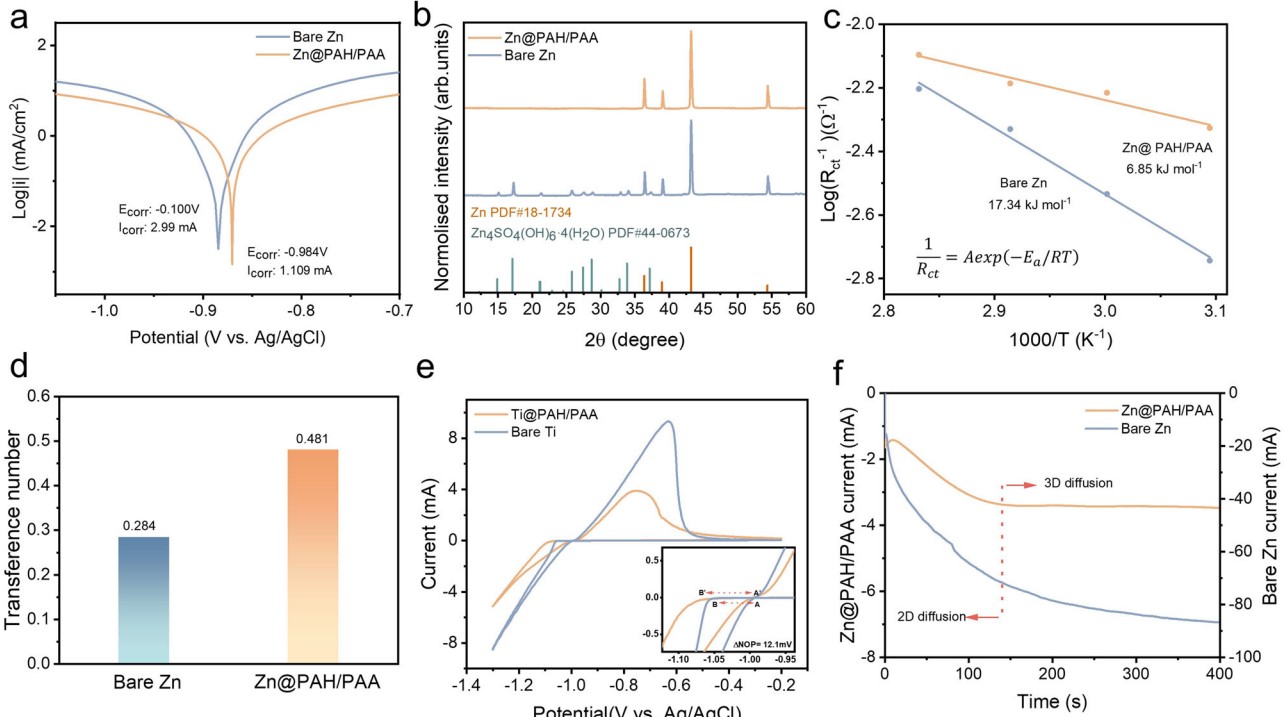

**Fig. 2 | Effects of the PAH/PAA multilayers on the diffusion and plating of Zn²⁺.** **a** Linear polarisation curves of Zn@PAH/PAA and Bare Zn. **b** XRD patterns of Zn@PAH/PAA and Bare Zn after 50 cycles at 0.5 mA cm⁻² and 0.5 mAh cm⁻². **c** Calculated activation energies for Zn@PAH/PAA and Bare Zn. **d** Zn transference numbers for Zn@PAH/PAA and Bare Zn. **e** Cycling voltammogram (CV) curves of Ti@PAH/PAA||Zn and Bare Ti||Zn asymmetric cells at 0.5 mV s⁻¹. **f** Chronoamperograms (CAs) of Zn@PAH/PAA and Bare Zn at an overpotential of −150 mV.

Coulombic efficiency of up to 99.8% after 1600 cycles at 0.5 mA cm⁻² and 0.25 mAh cm⁻² for the Cu||Zn asymmetric cell. The Zn||MnO₂ battery with the PAH/PAA coating layers displays a specific capacity of about 137 mAh g⁻¹ over 1000 cycles with 91.3% capacity retention at 2 A g⁻¹. Notably, although conventional research on the Zn anode has exhibited promising performance in coin cell tests, it is still far from the commercial application of AZIBs. Benefiting from the industrial maturity of the LbL self-assembly technology, we utilised practical equipment, such as extruders and coating machines, to fabricate the LbL anode. Through long cycling performance tests scaled up to large pouch cells, the effect of the LbL self-assembled polyelectrolytes on the practical applications of batteries was analysed in Zn||VO₂ pouch cells. The PAH/PAA multilayers enable a discharge capacity of 17.36 Ah over 250 cycles at 1.7 C. This work provides insight into the surface modification of Zn anodes, the design of the LbL self-assembled polyelectrolytes not only effectively enhances the electrochemical performance but also contributes to the environmental sustainability of the technology, making it a viable candidate for large-scale applications.

## Results

The LbL self-assembled PAH/PAA multilayers were prepared using the doctor-blading method[32–34]. By optimising the preparation process, three double PAH/PAA layers (Zn@PAH/PAA) with a thickness of ~280 nm can offer the most stable electrochemical performances (Figs. S1 and S3). The composition of Zn@PAH/PAA was successfully confirmed by Fourier Transform Infrared (FTIR) spectroscopy, as shown in Fig. S2. The bands at 1710 cm⁻¹ and 1247 cm⁻¹ are related to the C=O and C−O stretching vibration of carboxylic acid from the characterisation bands of PAA, respectively[35,36]. The bending of amine and amide groups from the characterisation bands of PAH are about 1633 cm⁻¹ and 1532 cm⁻¹ [36,37]. Compared with the bands of pure Zn@PAA and Zn@PAH, the significant band shifts for Zn@PAH/PAA

are observed, which are due to the electrostatic interactions between polyelectrolytes during the LbL self-assembly process. To investigate the effect of the LbL self-assembled PAH/PAA multilayers on Zn²⁺ plating/stripping behaviour, the thermostability and Zn²⁺ transport kinetics of Zn anodes were analysed. The linear polarisation curves indicate that the PAH/PAA multilayers can enhance the corrosion resistance of Zn anodes (Fig. 2a). The corrosion potential of Zn@PAH/PAA is increased from −0.1 V to −0.984 V, and the corrosion current is reduced to 1.109 mA, which is lower than that of bare Zn (2.99 mA). The higher corrosion potential and lower current mean more effective inhibition on the HER and by-products on Zn anodes[38]. To further verify the corrosion resistance of Zn@PAH/PAA, the HER on both Bare Zn and Zn@PAH/PAA electrodes within the initial 20 min at 30 mA cm⁻² was observed, shown in Fig. S4. After 10 min, small bubbles are generated and tend to accumulate on the Bare Zn electrodes. In contrast, no obvious bubbling appears on the Zn@PAH/PAA electrode. Moreover, the XRD pattern of Bare Zn after 50 cycles exhibits the strong diffraction peaks of the by-product (Zn₄SO₄(OH)₆·4H₂O, ZHS), which was not detected on the cycled Zn@PAH/PAA electrode (Fig. 2b). The results from SEM-EDS mapping in Fig. S5 also show a large amount of ZHS on the Bare Zn electrode after 50 cycles, compared with the Zn@PAH/PAA electrode. To further understand the mechanism of suppressing by-products, S 2*p* spectra for the cycled Zn@PAA/PAH electrode were characterised by X-ray photoelectron spectroscopy (XPS), as illustrated in Fig. S6. The detected peaks at approximately 168 eV and 162 eV correspond to SO₄²⁻ and ZnS contents, respectively[39,40]. After Ar⁺ etching, the signal of SO₄²⁻ becomes faint, while the signal of ZnS markedly increases. These results indicate that the PAH/PAA multilayers can significantly block SO₄²⁻ and suppress the HER and side reactions to enhance the thermostability of Zn anodes. To evaluate the ability of the LbL self-assembled polyelectrolyte coating to suppress side reactions compared to other hydrophilic polyelectrolyte coatings, a series of comparative experiments

with the sodium alginate (SA) and carboxymethyl cellulose (CMC) coatings were conducted. As shown in Fig. S7, Zn@PAH/PAA exhibits the relatively lowest magnitude compared to other monolayer hydrophilic polyelectrolytes SA and CMC. Moreover, XRD patterns reveal that compared to the SA coating, the PAH/PAA multilayers can more effectively suppress side reactions, as merely $Zn_4SO_4(OH)_6 \cdot 4H_2O$ by-product was detected. This result illustrates that although mono polyelectrolyte coatings can promote homogeneous Zn deposition by regulating $Zn^{2+}$ flux, their ability to repel $SO_4^{2-}$ ion is limited, especially under high current densities. In contrast, the ion-separation accelerating channels constructed by LbL self-assembled polyelectrolytes can better inhibit side reactions. In addition, ion-separation accelerating channels can modulate interfacial kinetics of $Zn^{2+}$ diffusion and deposition. As shown in Fig. S8, the PAH/PAA multilayers offer a strong hydrophilicity, of which the contact angle (58.6°) is smaller than that of Bare Zn (99.9°). The hydrophilicity of Zn@PAH/PAA can enable a lower interfacial energy barrier to regulate the diffusion of $Zn^{2+}$, as demonstrated in the activation energy analysis. Based on the EIS plots at different temperatures (Fig. S9), the interfacial activation energy ($E_a$) was evaluated through the Arrhenius equation (Fig. 2c). The hydrophilic PAH/PAA multilayers can reduce $E_a$ from 17.34 kJ mol$^{-1}$ to 6.85 kJ mol$^{-1}$, indicating that the PAH/PAA layers with a high zincophilicity can effectively regulate the $Zn^{2+}$ solvation structure and accelerate the transference. The Zn transference numbers of Zn@PAH/PAA and Bare Zn were calculated and shown in Fig. 2d and Fig. S10, where the high ionic conductivity of PAH/PAA multilayers can increase the Zn transference number from 0.284 to 0.481. To investigate the nucleation and growth behaviours of $Zn^{2+}$, the nucleation overpotential ($\eta$) on the Ti||Zn cell was evaluated (Fig. 2e). According to previous research[13,41], the critical nucleation radius ($\gamma_{crit}$) and nucleation rate ($\omega$) can be described as below:

$$\gamma_{crit} = h\sigma A/2\rho F\eta \tag{1}$$

$$\omega \propto \exp\left(\frac{-\pi LhA\sigma^2}{2\rho F\eta}\right) \tag{2}$$

Where $h$ is the height of the Zn atom, $\sigma$ is the interface tension, $A$ is the Zn atom mass, $\rho$ is the nucleus density, $F$ is the Faraday's constant, and $L$ is the Avogadro constant. As illustrated in Fig. 2e, $\eta$ is increased by 12.1 mV with the PAH/PAA multilayers, and the ratio of $\gamma_{crit}$ for Zn@PAH/PAA and Bare Zn is 0.47, attributed to an increased nucleation rate. Hence, a high nucleation overpotential can be attributed to a uniform and dense Zn deposition. Moreover, the chronoamperometry (CA) test reflects that $Zn^{2+}$ exhibits a 2D diffusion behaviour on Zn@PAH/PAA, compared with a 3D diffusion on the Bare Zn (Fig. 2f). The current change with time indicates the increase of effective Zn nucleation sites in chronoamperograms. The current for Zn@PAH/PAA remains stable after 140 s, whereas due to the aggregation of $Zn^{2+}$, the current for the Bare Zn continues to decrease within 400 s. Combined with the result of η, the PAH/PAA multilayers can regulate the Zn nucleation sites and make a uniform deposition to inhibit dendrite growth efficiently.

Since PAH/PAA multilayers can enable the 2D diffusion and plating of $Zn^{2+}$, the texture evolution and morphology of Zn anodes during cycling were further studied. XRD patterns of Zn@PAH/PAA under different cycles in Fig. 3a and Table S1 reveal that the (002) peak increases significantly after cycling. The intensity ratio $I_{(002)}/I_{(101)}$ becomes stronger, which is from 0.0611 (pristine Zn) to 0.1995 (15 cycles), 0.2298 (30 cycles), and 0.2973 (50 cycles). Density functional theory (DFT) calculations were carried out to analyse the adsorption energy of $Zn^{2+}$ and PAH$^+$/PAA$^-$ on Zn(002). PAH$^+$/PAA$^-$ exhibits a stronger adsorption energy (−0.760 eV) than $Zn^{2+}$ (−0.145 eV) on Zn(002), which is shown in Fig. 3b. Besides, the diffusion energy

barrier of $Zn^{2+}$ with the PAH/PAA multilayers coating on Zn(002) increases from 0.014 eV to 0.269 eV, suggesting that Zn@PAH/PAA can inhibit the aggregation of $Zn^{2+}$ and lead to a 2D diffusion and parallel plating (Fig. 3c)[42]. These results indicate that the PAH/PAA multilayers can induce the preferential nucleation and growth of $Zn^{2+}$ along Zn(002). SEM images show that with the increase of cycles, more and more (002) textures are observed on the Zn@PAH/PAA electrode (Fig. 3d). There are many horizontal (002) textures stacked together on the Zn@PAH/PAA after 50 cycles, where the thickness of the deposition layer is about 8 μm (Fig. S11). In sharp contrast, the morphology of Bare Zn after 50 cycles is uneven, with significant dendrite growth corresponding to a deposition layer of around 11 μm (Fig. S12). In-situ optical images and 3D depth profiles also verify that the PAH/PAA layers can guide a smooth and dense plating (Figs. S13 and S14). The aggregation and uneven nucleation of $Zn^{2+}$ on the Bare Zn electrode result in the dendrite formation and significantly roughen the surface after 5 min, while the surface of the Zn@PAH/PAA electrode remains smooth and homogeneous within 20 min. A solid electrolyte interphase (SEI) layer with a thickness of about 19 nm for the 50-cycled Zn@PAH/PAA electrode was observed through TEM images, shown in Fig. 3e. Further zooming in the SEI layer, the (002) textures are the primary plating orientation in the SEI layer, of which the area is larger than that of the (100) textures. Therefore, it can be known that the ion-separation accelerating channels formed by the LbL self-assembled PAH/PAA layers can induce the Zn nucleation and deposition along (002) lattice plane to form a smooth and dense Zn flake layer.

As the PAH/PAA multilayers can effectively enhance the thermostability and $Zn^{2+}$ transport kinetics to make the uniform Zn(002) deposition, the stability of Zn@PAH/PAA was discussed in relation to the symmetric Zn||Zn cell. The Zn@PAH/PAA electrode exhibits a stability around 1100 h at 1 mA cm$^{-2}$ and 1 mAh cm$^{-2}$, whereas the Bare Zn electrode suffers a short circuit around 76 h (Fig. 4a). Moreover, the stability of Zn@PAH and Zn@PAA electrodes at 1 mA cm$^{-2}$ and 1 mAh cm$^{-2}$ were also examined to better analyse the impact of PAH/PAA multilayers compared to PAH or PAA monolayers on battery performance (Fig. S21). Compared to the long stability of the Zn@PAH/PAA electrode, Zn@PAH and Zn@PAA electrodes experience battery failure within 200 h. This once again proves that the PAH and PAA monolayers cannot provide the same level of performance due to their respective limitations when applied individually, while the PAH/PAA multilayers combine the advantages of PAH (ionic conductivity and adhesion) and PAA (repulsion of $SO_4^{2-}$ and mechanical stability) to offer a synergistic effect that ensures long-term cycling reversibility. With increasing the current and capacity densities to 5 mA cm$^{-2}$ and 5 mAh cm$^{-2}$, the Zn@PAH/PAA electrode (340 h) still shows a more extended cycling performance than the Bare Zn electrode (160 h), illustrated in Fig. 4b. Furthermore, the Zn@PAH/PAA electrode presents a high depth of discharge (DOD): 53.4% within around 170 h cycling at 8 mA cm$^{-2}$ and 4 mAh cm$^{-2}$ (Fig. 4c). As shown in Fig. S15, the voltage profile for the Zn@PAH/PAA electrode at different current densities and capacities exhibits a larger potential difference than that of the Bare Zn electrode, which is due to the lower nucleation radius and higher nucleation rate of Zn@PAH/PAA as mentioned in Fig. 2e. In addition, the large voltage difference of the Zn@PAH/PAA electrode may also be caused by the PAH/PAA layers inducing the orientational plating of $Zn^{2+}$ along Zn(002). Compared with the Bare Zn electrode, the rate performance of the Zn@PAH/PAA exhibits a $Zn^{2+}$ plating/stripping stability at various current densities from 1 mA cm$^{-2}$ to 10 mA cm$^{-2}$ at 1 mAh cm$^{-2}$ (Fig. S16). These GCD tests indicate that the PAH/PAA multilayers enable high stability and reversibility for the Zn anode. The Coulombic efficiency (CE) was analysed by the asymmetric Cu||Zn cell. As shown in Fig. 4d, e, the initial CE of Cu@PAH/PAA (92.3%) is higher than that of bare Cu (86.7%), and the Cu@PAH/PAA electrode offers a CE of around 99.8% after 1600 cycles. In contrast, the bare Cu electrode suffers a rapid decline in CE after about 70 cycles.

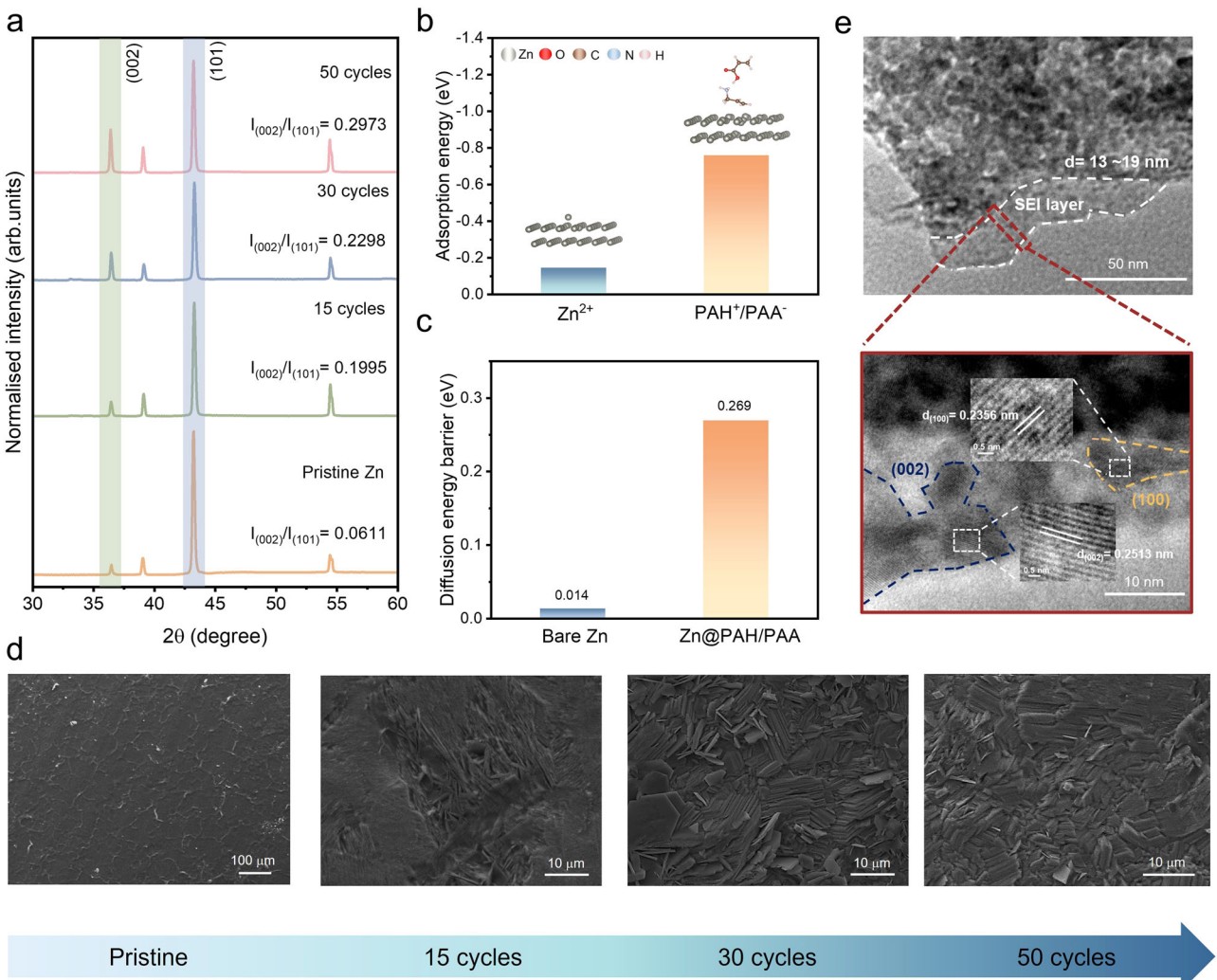

**Fig. 3 | Surface texture characterisations for Zn²⁺ plating. a** XRD patterns of Zn@PAH/PAA under different cycles. **b** Adsorption energies of Zn²⁺ and PAH⁺/PAA⁻ on the Zn (002) lattice plane. **c** Diffusion energy barriers of Zn²⁺ on Zn (002) crystal plane with/without PAH/PAA multilayers. **d** SEM images under different cycles. **e** TEM images of Zn@PAH/PAA after 50 cycles. (all characterisations cycled at 0.5 mA cm⁻² and 0.5 mAh cm⁻²).

The CE performance indicates that the PAH/PAA multilayers can efficiently inhibit the side reactions and passivation of Zn anodes. Hence, the initial nucleation overpotential of Cu@PAH/PAA increases from 0.0746 V to 0.0961 V, which once again approves that PAH/PAA layers can enable a lower nucleation radius and a higher nucleation rate. The cumulative plated capacity (CPC) of the Cu@PAH/PAA electrode is 396 mAh cm⁻², which is more competitive than most recent research on the surface modification of Zn anodes (Fig. 4f and Table S2)[43–48].

In-situ Raman spectra were recorded during each cycle to investigate the mechanism of Zn²⁺ plating/stripping behaviours on Zn@PAH/PAA. As shown in Fig. S17, the band assigned to the -CH₂ stretching vibration is at 2928 cm⁻¹ for Zn@PAH/PAA, while it is at about 2930 cm⁻¹ for Zn@PAH[49]. This difference is due to the electrostatic interactions between PAH and PAA polyelectrolytes as confirmed by FTIR. After immersing Zn@PAH/PAA to 2 M ZnSO₄ for 15 min, the -CH₂ stretching vibration band moves to 2931 cm⁻¹, indicating the ionic interaction between −CH₂−NH₃⁺ and SO₄²⁻. Moreover, the band shape between 1400 cm⁻¹ and 1450 cm⁻¹ is related to the −RNH₃⁺ deformation and -CH₂ bending, by which the shape change further confirms the interaction between -CH₂-NH₃⁺ and SO₄²⁻ [50,51]. The band from 1750 cm⁻¹ to 1600 cm⁻¹ for Zn@PAH/PAA corresponds to the vibration of symmetric C−H of PAH and C=O in carboxylate groups of PAA, while it splits into two bands after immersing with 2 M ZnSO₄,

owing to the ionic interaction between −COO⁻ and Zn²⁺ that enhances the intensity of C=O band[52,53]. The periodic band changes can be observed in each plating/stripping cycle, as illustrated in Fig. 5a and Table S3. The ionic interaction between SO₄²⁻ and −CH₂−NH₃⁺ shifts the band of −CH₂ vibration to a lower wavenumber during Zn²⁺ plating and a higher wavenumber during Zn²⁺ stripping. Correspondingly, the relative band intensity of −CH₂ bending and NH₃⁺ deformation changes between 1400 cm⁻¹ and 1450 cm⁻¹. Moreover, the coordination of Zn²⁺ and −COO⁻ makes the band of C=O vibration move to a lower wavenumber during plating, while the band moves to a higher wavenumber due to the escape of Zn²⁺ during stripping. These regular and reversible band changes indicate the formation of dual-ion channels between PAH and PAA polyelectrolytes. The binding energy calculations reveal that PAA⁻ coordinates with Zn²⁺ to form [PAA⁻−Zn(H₂O)₄]⁺, regulating the solvation structure of Zn²⁺. Additionally, PAH⁺ binds with SO₄²⁻ to form the stable [(PAH⁺)₃ Zn(SO₄)₂]⁺ coordination structure (Fig. 5b, c). Based on the above results, the interfacial mechanism during Zn²⁺ plating/stripping on the Zn@PAH/PAA electrode is illustrated in Fig. 5d. The ion-separation accelerating channels in the structure of PAH⁺ − SO₄²⁻ − Zn²⁺ − PAA⁻ are constructed by the LbL self-assembled PAH/PAA multilayers, where PAA⁻ regulates the Zn²⁺ solvation shell and accelerates Zn²⁺ transport at the inner Helmholtz plane, and PAH⁺ binds with SO₄²⁻ to inhibit the formation of ZHS. Indeed, the PAH/PAA

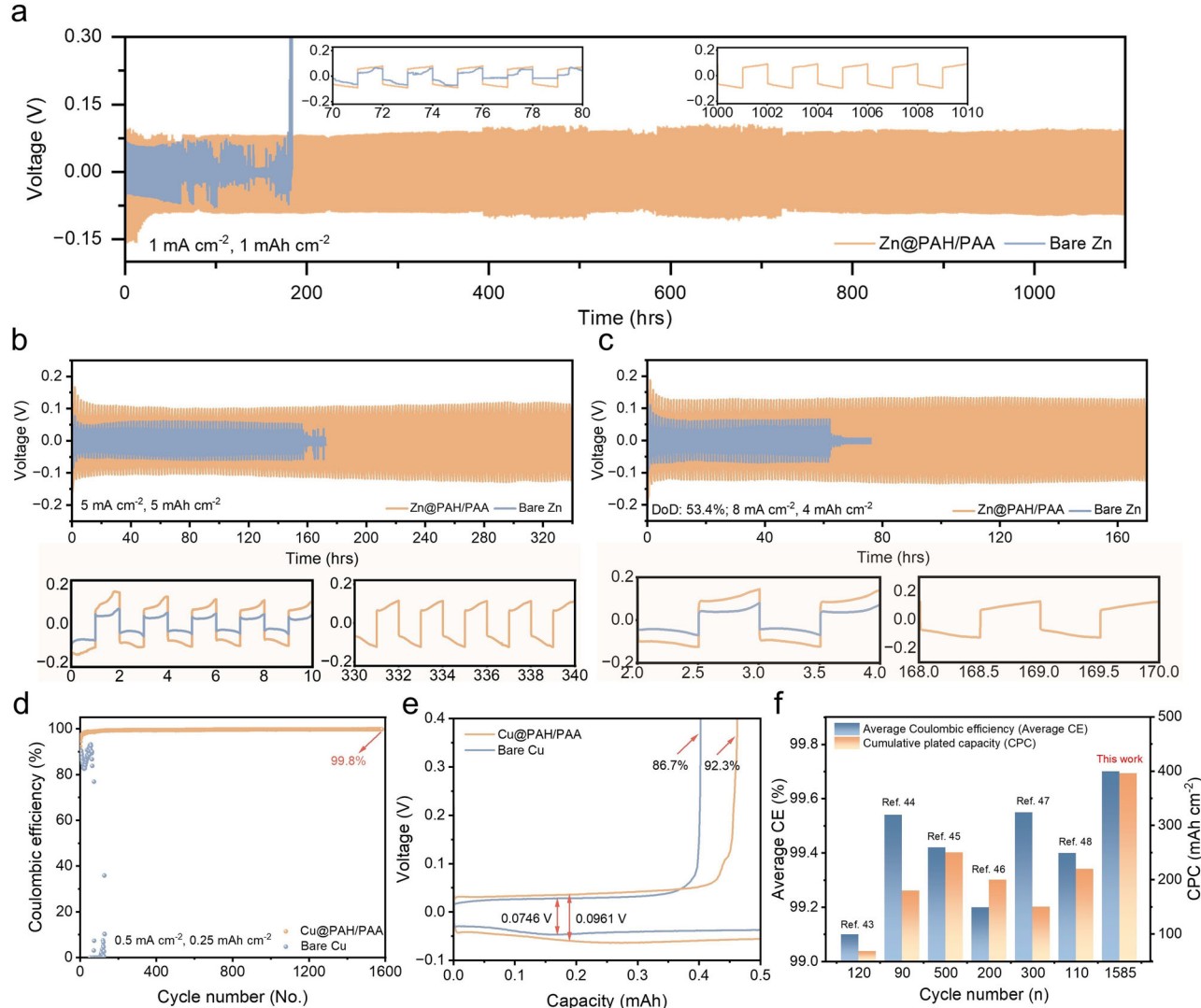

**Fig. 4 | Electrochemical performances of the symmetric and asymmetric cells with the coating of PAH/PAA multilayers.** The galvanostatic cycling performances of Zn||Zn symmetric cells at **a** 1 mA cm⁻² and 1 mAh cm⁻², **b** 5 mA cm⁻² and 5 mAh cm⁻², and **c** 8 mA cm⁻² and 4 mAh cm⁻². **d** The CE performance of Cu||Zn asymmetric cells at 0.5 mA cm⁻² and 0.25 mAh cm⁻². **e** The corresponding voltage profile at first cycle of Cu||Zn asymmetric cells at 0.5 mA cm⁻² and 0.25 mAh cm⁻². **f** Comparison of recent anode performance regarding CPC, cycle number, and average CE.

multilayers also induce the Zn nucleation and deposition along (002) texture to form the uniform and dense Zn flake layer, thereby suppressing dendrite formation.

A Zn||commercial $MnO_2$ battery was assembled to investigate the effect of the PAH/PAA multilayers on the electrochemical performance of the full cell. The CV curves at a scan rate of 0.1 mV s⁻¹ are shown in Fig. S18, where two redox peaks are related to the intercalation and de-intercalation of $Zn^{2+}$ and $H^+$, respectively[54]. Because of the large nucleation overpotential and high nucleation rate of Zn@PAH/PAA mentioned earlier, the polarisation for the Zn@PAH/PAA battery is more significant than that of the Bare Zn battery. The rate performance for both electrodes from 0.1 A g⁻¹ to 5 A g⁻¹ is illustrated in Fig. 6a and Fig. S19. Due to the improved interfacial kinetics and thermostability of Zn anodes, the Zn@PAH/PAA battery exhibits a higher specific capacity and reversibility than the Bare Zn battery at each current density (262 mAh g⁻¹ at 0.1 A g⁻¹ and 101 mAh g⁻¹ at 5 A g⁻¹). Furthermore, the Zn@PAH/PAA battery displays a specific capacity of -137 mAh g⁻¹ over 1000 cycles with 91.3% capacity retention at 2 A g⁻¹. In contrast, the Bare Zn battery suffers a rapid capacity decline of 73 mAh g⁻¹ after 620 cycles (Fig. 6b). These results indicate that the LbL self-assembled PAH/PAA multilayers can significantly inhibit the side reactions and

enhance the CE value, thereby improving the overall performance of the batteries. We also further investigated the LbL self-assembly technique in promoting the practical application of AZIBs. To assess performance under high current density and mass loading, Zn@CMC and Zn@PAH/PAA electrodes in a 15 Ah full cell were analysed to evaluate actual anode behaviour. As illustrated in Fig. S20, Zn@PAH/PAA maintains stable GCD performance above 15 Ah with nearly 100% CE value over 200 cycles at a charging rate of almost 2 C. In contrast, Zn@CMC exhibits a drop of around 85 cycles, followed by a rapid capacity loss. The inset shows post-analysis, revealing delamination between the CMC coating and the Zn anode. This delamination leads to wrinkling in the separator and cathode, causing a sharp decline in CE value. Since side reactions are not well suppressed, the monolayer coating easily delaminates. In contrast, the PAH/PAA multilayers effectively suppress side reactions, allowing for a stable and robust plating regime and improving battery life under high charging rates. These results illustrate the enhanced performance of the PAH/PAA multilayers compared to the mono hydrophilic polyelectrolyte coatings, particularly in terms of maintaining structural integrity and CE value over extended cycling. Moreover, as shown in Fig. 6c, the Zn@PAH/PAA||commercial $VO_2$ pouch cell with the high mass loading

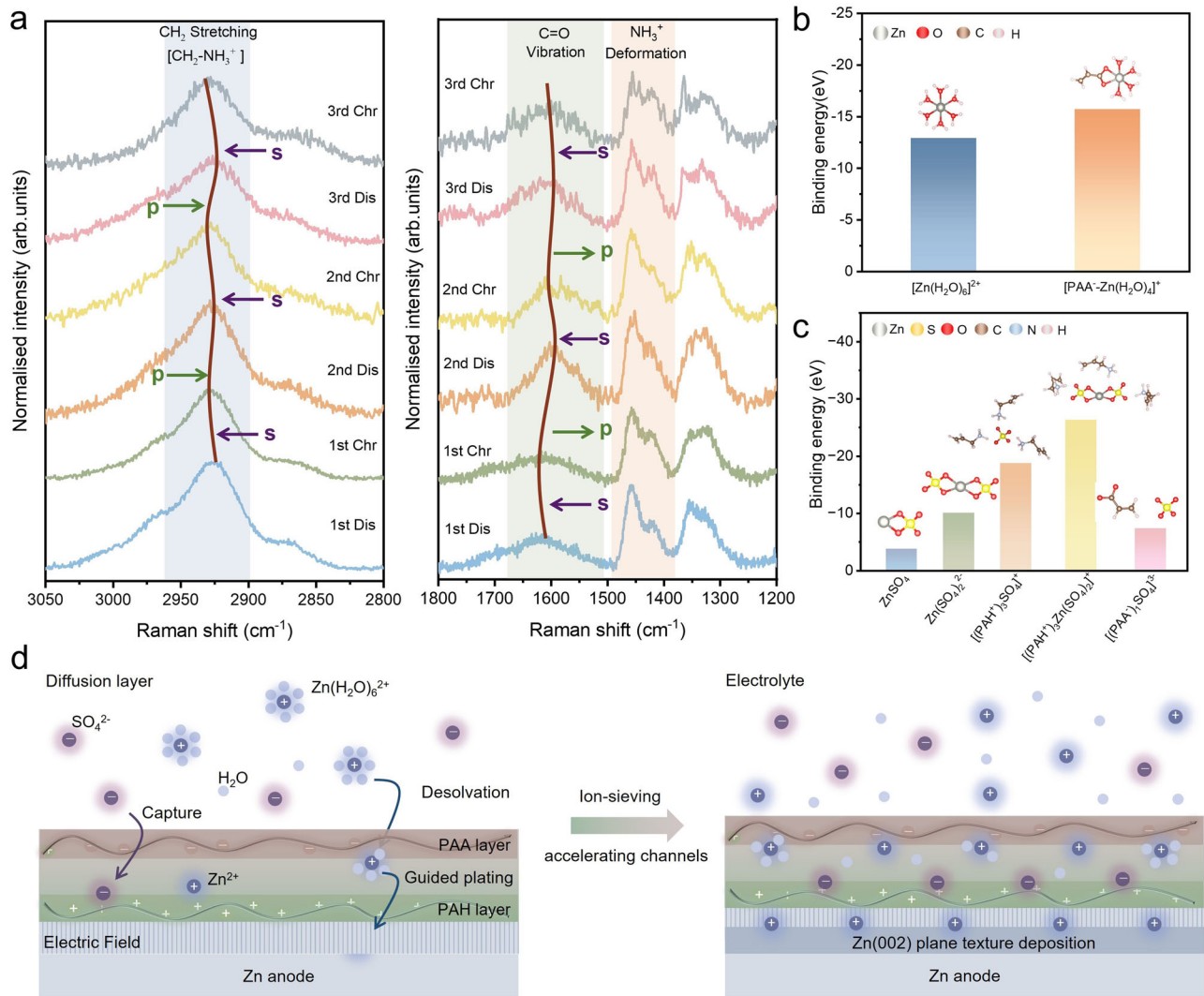

**Fig. 5 | Mechanism of the Zn²⁺ plating/stripping behaviours on Zn@PAH/PAA.**
**a** In-situ Raman spectra of the Zn@PAH/PAA electrode at 35 mA cm⁻² for 3600 s each cycle. (s: stripping, p: plating). **b** The binding energy of different coordination

structures of PAA⁻. **c** The binding energy of different coordination structures of PAH⁺. **d** Schematic diagram of the Zn deposition process on Zn@PAH/PAA.

(>8 mg cm⁻²) was assembled, which exhibits an Ah-level residual discharge capacity of 17.36 Ah after 250 cycles at 1.7 C, with a capacity retention of 96.3%. The capacity and high specific current achieved in this work are much higher than most previous work on Zn metal anodes, indicating the effect and potential of the LbL self-assembly technique to improve the anode stability in the practical application of Zn-ion batteries (Fig. 6d and Table S5)[55–59].

## Discussion

In summary, the LbL self-assembled PAH/PAA multilayers with high mechanical strength and ionic conductivity can effectively enhance the reversibility and stability of the Zn anode. The ion-separation accelerating channels constructed by the multilayers not only enable a high zincophilicity to regulate the Zn²⁺ desolvation process but also capture $SO_4^{2-}$ to suppress the formation of by-products. Moreover, the PAH/PAA layers can induce Zn deposition along the (002) crystal plane to form a uniform and dense layer, inhibiting dendrite formation. Since the PAH/PAA multilayers can enhance the interfacial Zn²⁺ transport kinetics and thermostability, the Zn‖Zn symmetric cell achieves a long stability over 1100 h at 1 mA cm⁻² and 1 mAh cm⁻², and the Cu‖Zn asymmetric cell exhibits a Coulombic efficiency of 99.8% and a high CPC of 396 mAh cm⁻² after 1600 cycles at 0.5 mA cm⁻² and

0.25 mAh cm⁻². Moreover, the PAA/PAH multilayers enable the Zn‖VO₂ pouch cell to retain a high discharge capacity of 17.36 Ah after 250 cycles at 1.7 C with a high mass loading. We anticipate that this work inspires a strategy for constructing ion-separation accelerating channels through the LbL self-assembly of polyelectrolytes to protect the metal anode, promoting practical applications of aqueous rechargeable batteries.

## Methods

### Preparation of LBL self-assembly of PAH/PAA multilayers coated Zn anode

500 mg PAH (Sigma-Aldrich, average Mw ~50,000) was mixed in 10 mL distilled water to form PAH aqueous solution. PAA aqueous solution was prepared with 450 mg PAA (Sigma-Aldrich, average $M_w$ ~450,000) and 10 mL distilled water. Zn@PAH/PAA was coated by a 50 μm doctor blade, where the PAH layer was coated on the Zn foil first and dried under UV light; after rinsing with water for 5–10 min, the PAA layer was then coated on the Zn foil (50 μm, Hefei Wenzhou Co., Ltd). Repeat the above steps to obtain the PAH/PAA multilayers coated Zn foil. According to GCD curves (Fig. S1), the three double-layer coated Zn foil exhibits the best cycling performance. Therefore, three double PAH/PAA multilayers were used to prepare the Zn@PAH/PAA in this work.

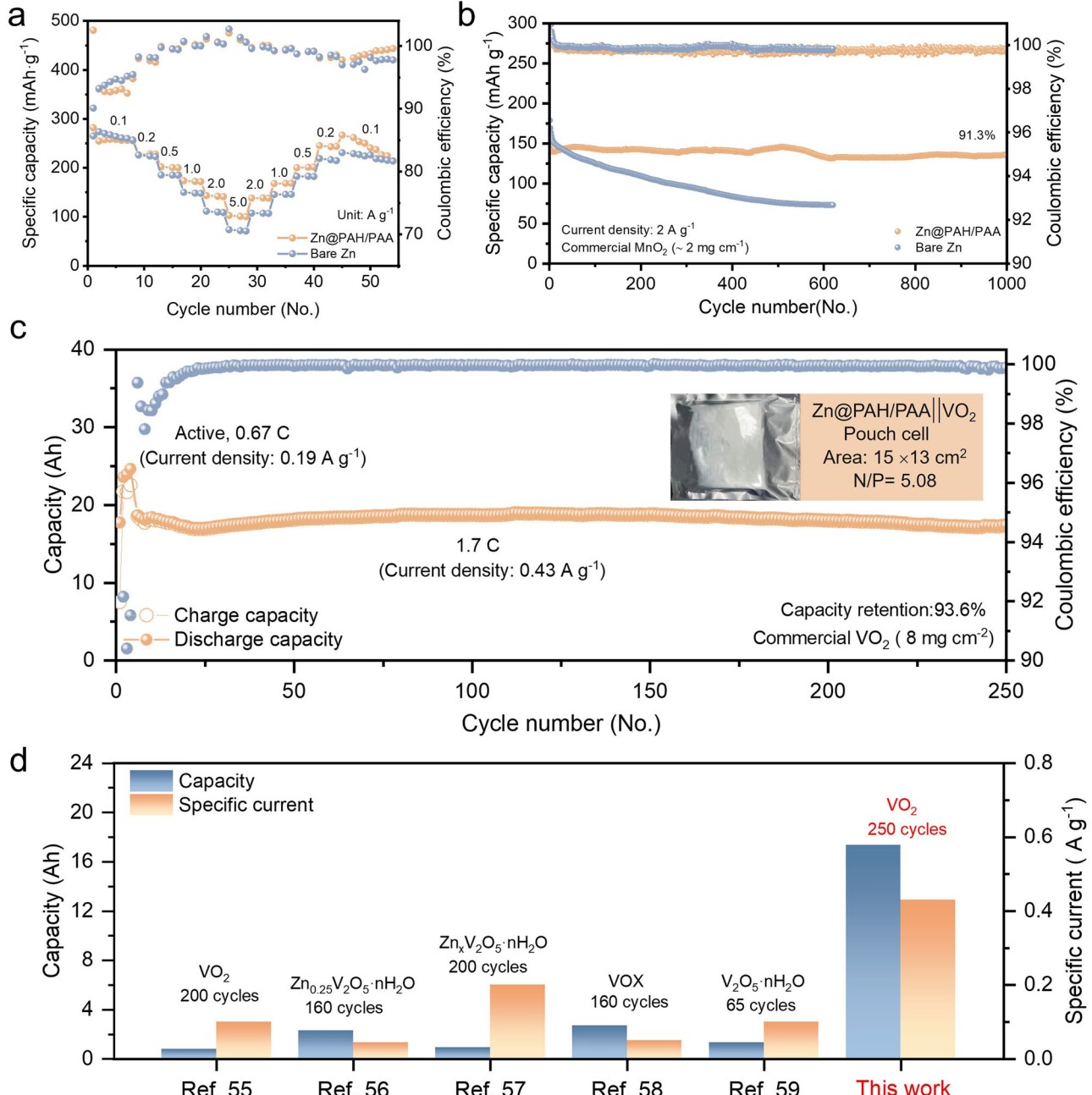

**Fig. 6 | Electrochemical performances of the Zn@PAH/PAA full cell. a** The rate performance of Zn‖MnO₂ coin cell at different current densities of 0.1, 0.2, 0.5, 1, 2, 5 A g⁻¹. **b** Long-term cycling performance of Zn‖MnO₂ coin cell at the current density of 2 A g⁻¹. **c** Long-term cycling performance of Zn‖VO₂ pouch cell at 1.7 C (Optical image of the Zn‖VO₂ pouch cell). **d** Comparison of recent anode performance regarding Zn metal pouch cells on capacity, cycle number, and specific current.

## Battery assembly

For full cell assembly, commercial MnO₂ (Sigma-Aldrich) was mixed and grounded with carbon black and PVDF at a 7:2:1 weight ratio in NMP solution. The mixture was evenly coated on a hydrophilic carbon paper with the areal loading mass around 1.5–2.5 mg cm⁻², and dried in a vacuum oven at 65 °C overnight. Afterward, the dried cathode was assembled with the PAH/PAA multilayers coated Zn anode in a CR2032-coin cell using 2 M ZnSO₄ and 0.2 M MnSO₄ as the electrolyte. The N/P ratio is around 9.50. For symmetrical cell assembly, the PAH/PAA multilayers coated Zn foil was assembled with 2 M ZnSO₄ as the electrolyte. Glass fibre (Whatman GF/D) was applied as the separator. Electrolyte solutions were prepared a day before the cell assembly and stored in the dry cabinet (25 °C, relative humidity 75%). For the Cu‖Zn

and Ti‖Zn tests, Cu foil (25 μm, MTI Corp) and Ti foil (50 μm, Hebei Xinji Co., Ltd) were employed.

## Fabrication of pouch cells

For the pouch cells, VO₂ (Zhejiang Vastech Co., Ltd) was used for the demonstration. The cathode was fabricated by mixing VO₂ with carbon black and PVDF at an 8:1:1 weight ratio in an NMP solution. The mixture was evenly coated on the stainless-steel foil (50 μm, Zhejiang Vastech Co., Ltd) with the areal loading mass around 8 mg cm⁻² by extrusion coater and dried in a vacuum oven at 65 °C overnight. Afterward, the dried cathode was cut into a rectangular shape in 15 × 13 cm² with a thickness of 230 μm. Afterward, the dried cathode was assembled with the PAH/PAA multilayers coated Zn anode, using 3 M Zn(CF₃SO₃)₂ (Sigma-Aldrich). The

coated Zn anode was fabricated in the same approach as mentioned above, where the size is $15 \times 13\ cm^2$ with a thickness of 30 μm. Glass fibre (150 μm, Chongqing Ouleji Co., Ltd) was used as the separator. Around 30 layers of cathodes were laminated and welded in an aluminium bag. The N/P ratio is around 5.08. Zhejiang Vastech Co., Ltd, conducted the sealing procedure and welding process. The obtained pouch cell was rested for six hours before the electrochemical test.

## Materials characterisation

X-ray diffraction (XRD) patterns were obtained by a Bruker Vantec500 under the radiation source of Cu metal. Scanning electron microscopy (SEM) images were collected by a JEOL JSM-6701F Field Emission Scanning Electron Microscope (JEOL, Japan) at the acceleration voltage of 15 kV. The Energy-dispersive X-ray spectroscopy (EDX) images were collected by a Carl Zeiss EVO MA10 (Carl Zeiss AG, Germany) and Ultim Extreme Silicon Drift Detectors (Oxford Instrumental plc, UK). Transmission electron microscope (TEM) images were carried out by a JEOL JEM-2100 Electron Microscope. Fourier transform infrared spectroscopy (FTIR) was measured by a Shimadzu IRTracer-100 with the wavenumber from 400 to 4000 $cm^{-1}$. The Raman data was obtained by a Thermo Scientific™ DXR3 Raman Microscope with a laser wavelength of 532 nm. The optical images and depth profile were collected by a Keyence VXH-7000N Digital Microscope; the HER reaction was observed on the anode and Zn deposition was observed on the cathode in the discharge process.

## Electrochemical characterisation

The long-term galvanostatic charge-discharge (GCD) test was operated by NEWARE battery testing systems (CT-4000-5V 10 mA, Shenzhen, China). The cyclic voltammetry (CV) was measured by a VMP3 Biologic potentiostat. The electrochemical impedance spectroscopy (EIS) was tested by a VMP3 Biologic potentiostat between $10^{-2}$ and $10^5$ Hz. Potentiostatic mode was applied and recorded at 7 points/decade, and 10 s was applied during open-circuit potential before measurement. The electrochemical characterisations of the Zn||$MnO_2$ and Zn||$VO_2$ cells were conducted at the voltage window 0.8–1.9 V and 0.3–1.7 V, respectively. All electrochemical tests were carried out at a temperature of 25 °C ± 2 °C. For the pouch cell test, the cell is placed in the battery testing room controlled by an automatic air conditioner. For the full cell test in the coin cell structure, we have conducted at least 3 times for each electrochemical characterisation. For the full cell test in the pouch cell structure, we have conducted 2 times to verify the performance under the high mass loading.

Capacity retentions were calculated based on the discharged capacity at the certain cycle mentioned in the manuscript, where 1000 cycles were selected. The specific current is the applied current over the active materials in the cathode expressed as A $g^{-1}$.

The specific capacity (mAh $g^{-1}$) is calculated based on the discharged capacity (mAh) per unit mass (g) of the active material using the equation below:

$$C_{specific} = \frac{Q}{m} \tag{3}$$

where $Q$ is the total discharged capacity and $m$ is the mass of the active material.

The term C-rate refers to the discharge or charge current, in amperes, expressed as a multiple of the rated capacity in ampere-hours. 1 C-rate means the cell is charged or discharged at a current that would fully charge or discharge the cell in one hour. A 0.5 C-rate would correspond to charging or discharging the cell in two hours, and a 2 C-rate would correspond to charging or discharging the cell in half an hour.

## Computational details

Density functional theory (DFT) calculations were performed using the Vienna ab initio Simulation Program (VASP). The generalised gradient approximation (GGA) method in the Perdew-Burke-Ernzerhof (PBE) functional was applied to describe the exchange-correlation interaction. A conjugate gradient algorithm was employed for geometrical optimisation. The convergence criterion for the total energy and ionic force were $10^{-4}$ eV and 0.03 eV/Å, respectively. The cut-off energy for the plane-wave basis set was 500 eV. Monkhorst-Pack scheme was used to sample the Brillouin zone with a $k$-point of $1 \times 1 \times 1$ for geometrical optimisation. The van der Waals (vdW) interaction was considered through DFT-D3 correction. The vacuum layer was larger than 20 Å to avoid the interlayer interactions. To obtain the diffusion energy barriers of Zn ions on the electrode surface, the climbing image nudged elastic band (CI-NEB) method was adopted. A $6 \times 6$ supercell of Zn (002) surface containing 2 atomic layers was constructed as the Zn electrode model.

The adsorption energy ($E_{ads}$) is defined as

$$E_{ads} = E_{total} - E_{surface} - E_{adsorbent} \tag{4}$$

Where $E_{total}$, $E_{surface}$, and $E_{adsorbent}$ represent the total energies of the adsorption system, the substrate, and the adsorbent, respectively. It means stronger adsorption with more negative adsorption energy.

The binding energy is calculated as

$$E_b = E_{total} - \sum_i n_i \mu_i \tag{5}$$

Where $n_i$ is the number of atoms, and $\mu_i$ is the corresponding chemical potential.

## Data availability

The data support in this study are available within the Supplementary information and Source data. Source data have been deposited in the Figshare Database (https://doi.org/10.6084/m9.figshare.28237529.v2).

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

## Acknowledgements

The authors would like to thank the support from Fundamental Research Funds for the Central Universities (x2wjD2240360) received by H.D., Engineering and Physical Sciences Research Council (EPSRC, EP/V027433/3) received by G.H., and UK Research and Innovation (UKRI) under the UK government's Horizon Europe funding (101077226; EP/Y008707/1) received by G.H. Especially thanks to the Science and Technology Facilities Council Early Research Award for financial support (ST/R006873/1) received by H.D. and the support given to Vastech battery company for pouch cell fabrication. X.H. would like to thank the funding support from China Scholarship Council/University College London for the joint Ph.D. scholarship.

## Author contributions

H.D., X.H. and G.H. conceived the project. X.H. and H.D. conducted the experiments, analysed the data, and wrote the manuscript. T.W. operated the TEM characterisation. N.G. performed the DFT calculations. H.H. conducted the contact angle test. H.D., X.G., Y.D., Y.L. and D.B. helped with the pouch cell test. H.D., G.H. and I.P.P. contributed to the discussion of results and revised the manuscript. The manuscript was written through contributions of all authors.

## Competing interests

The authors declare no competing interests.
