## [Transparent Peer Review file · Nature Communications]

Self-assembled polyelectrolytes with ion-separation accelerating channels for highly stable Zn-ion batteries

Corresponding Author: Professor Guanjie He

Version 0:

Reviewer comments:

Reviewer #1

(Remarks to the Author)

This work reported the use of two polyelectrolytes with opposite charges to form a self-assembly layer. The authors claimed that this self-assembly layer acts as an ion-sieving channel to stabilize the zinc anode for a zinc-ion battery. Currently, there are many similar works published using polyelectrolytes or polymer additives to stabilize the zinc anode. For example: ACS Nano 2024, 18, 6, 5003–5016; J. Alloys Comp. 2023, 950, 169836; Angew. Chem. Int. Ed. 2023, 62, e202311268; Nat. Commun. 2023, 14, 6526.; Energy Storage Materials 2022, 45, 1084-1091. The difference is this work using two polymer electrolytes. And I could not agree that this assembly can enable an ion-sieving channel, which may exaggerate. Two different charged polymers will naturally attract oppositely charged ions. However, it does not imply an ion channel, which is often achieved by metal-organic frameworks or covalent-organic frameworks. Nevertheless, this is good work showing incremental advances in optimizing the utilization of polymer electrolytes for zinc-ion batteries and demonstrating the pouch battery application, but not enough novelty and general interest for Nature Communication. The work is more suitable for an energy-related journal.

Reviewer #2

(Remarks to the Author)

In this work, the authors report a method to form polyelectrolyte layer to promote uniform Zn diffusion and deposition on the anode for Zinc-ion batteries. The concept of the polyelectrolyte layer formation via LbL is not new, but the authors make an interesting claim that the layer acts as an ion-sieving accelerating channel, which remains to be further validated. The reported performance of the ZIB is impressive and the authors clearly know the current challenges in the subject. However, the PAH/PAA chemistry has been around for a long time, and the LbL method is also a well-established technique. Although this is an interesting work with good results, in the Reviewer's opinion, this manuscript does not seem to provide "new" chemistry nor innovative method to be considered in the Nature communication journal.

- Can authors please provide direct proof that the PAH/PAA layer acts as an ion-sieving "accelerating" channel? How does it accelerate the Zn ion flux? It has been well-reported that simply making the surface hydrophilic improves the performance. The change in the interfacial activation energy shown in this work could have been obtained by other hydrophilic modification. What is different and unique about PAH/PAA compared to other hydrophilic modification works? The authors should provide a convincing control experiment where another hydrophilic modification is used with similar surface properties.

- The preferential nucleation of Zn²⁺ along Zn(002) surface is very interesting. Has this been observed before? Do the authors expect the same outcome on different hydrophilic modification methods? Again, similarly to my previous comment, what is unique about PAH/PAA chemistry that leads to preferential nucleation? I think the DFT works should be moved into the main manuscript, if it can provide convincing evidence.

Version 1:

Reviewer comments:

Reviewer #2

(Remarks to the Author)

The authors have adequately responded to the Reviewer's comments, and it is very much appreciated. I still think that the work does not show novelty in terms of chemistry nor methods (LbL), and another concern is that the resulting data seem like a gradual improvement compared to the other literature on the same topic (surface coatings). Nevertheless, the presented data are certainly interesting to the researchers in the topic, and backed by sufficient data, and the authors have shown its efficacy in relatively large scale pouch module. My suggestion is to either accept the manuscript (if the Editor seems fit) or transfer it to another specific.

Reviewer #3

(Remarks to the Author)

The authors have provided a comprehensive study on the development of a PAH/PAA multilayer coating for Zn anodes in aqueous Zn batteries. The novelty of this work lies in the innovative approach to enhancing the performance of zinc (Zn) anodes in aqueous zinc-ion batteries by employing layer-by-layer (LBL) self-assembly of poly(allylamine hydrochloride) (PAH) and poly(acrylic acid) (PAA) multilayers. The manuscript effectively addresses the key issues and provides substantial experimental and computational support for the claims made in the revised manuscript.

Reading through the comments and response, I think the authors have well answered the critical questions posed in the review process. This work satisfied the standard of Nature Communications. I would like to raise a few minor points that could be addressed in the final revision:

1. While the long-term cycling performance is impressive, could the authors provide more detailed insights into the mechanisms driving degradation, if capacity-voltage plots can be provided.
2. Have the authors considered whether this PAH/PAA coating method would be compatible with industrial application? Any price related calculation could be provided.
3. Can the authors elaborate on how the specific combination of PAH and PAA multilayers improves zinc anode performance compared to using either PAH or PAA alone?

Version 2:

Reviewer comments:

Reviewer #3

(Remarks to the Author)

The paper has been thoroughly refined and improved, and it is now ready for publication in its present state. The enhancements have addressed all previous concerns.

Manuscript Title: Self-assembled polyelectrolytes with ion-sieving accelerating channels for highly stable Zn-ion batteries

Manuscript ID: NCOMMS-24-37366-T

Comments:

Reviewer 1

This work reported the use of two polyelectrolytes with opposite charges to form a self-assembly layer. The authors claimed that this self-assembly layer acts as an ion-sieving channel to stabilize the zinc anode for a zinc-ion battery.

Response: Thanks for your comment. We have added extra experiential results and changed the explanation accordingly. We expect that the revised version will satisfy the requirement from the referee.

1. Currently, there are many similar works published using polyelectrolytes or polymer additives to stabilize the zinc anode. For example: ACS Nano 2024, 18, 6, 5003–5016; J. Alloys Comp. 2023, 950, 169836; Angew. Chem. Int. Ed. 2023, 62, e202311268; Nat. Commun. 2023, 14, 6526.; Energy Storage Materials 2022, 45, 1084-1091. The difference is this work using two polymer electrolytes.

Response: Thank you for your insightful comments. Indeed, several studies have been published on the use of polyelectrolytes to stabilize the Zn anode. However, most of these works focus on using a single polyelectrolyte to protect the anode interface. As you mentioned in the comment, our previous work (Angew. Chem. Int. Ed. 2023, 62, e202311268) utilized the polyanionic electrolyte sodium alginate (SA) to enhance the performance of the Zn anode. While this approach showed promise, we observed certain limitations. Specifically, a mono-polyelectrolyte interface has restricted control over ion diffusion and provides limited mechanical strength to the anode surface.

We have conducted additional tests to clarify the difference. As shown in Figure S7 below, although there is no huge difference regarding the plating overpotential, Zn@PAH/PAA still exhibits the relatively lowest magnitude compared to other monolayer polyelectrolytes SA and CMC. Although anionic polyelectrolytes can promote homogeneous Zn²⁺ deposition by regulating Zn²⁺ flux, they show a limited ability to repel SO₄²⁻, particularly under high current densities. This can lead to the formation of by-products. By comparing the XRD patterns between cycled Zn@PAH/PAA and Zn@SA, we can clearly see that the PAH/PAA multilayers can more effectively suppress side reactions, as merely Zn₄SO₄(OH)₆·4H₂O byproduct was detected. Our approach creates dual channels that simultaneously regulate the diffusion of both Zn²⁺ and SO₄²⁻ ions. The strong electrostatic interactions between the carboxylate anions (COO⁻) in PAA and the ammonium cations (NH₃⁺) in PAH create a separation mechanism that block SO₄²⁻, preventing their participation in undesirable side reactions. This capture mitigates the

formation of insulating layers of by-products, which are a common challenge in conventional systems using mono-polyelectrolytes. As shown in Figure 3c, the Zn||Zn performance is still stable under a high DoD of 53.4%. The nucleation radius for Zn@PAH/PAA and Bare Zn is 0.47, which is attributed to an increased nucleation rate and a uniform plating.

Figure S7. (a) Optical image after coating in mono and multilayers. (b) Zn||Zn performance. (c) XRD comparisons after 50 cycles at 1 mA cm⁻² and 1 mAh cm⁻².

In addition, while conventional coating methods have shown good results in coin cell tests, they are often not feasible for large-scale applications. In contrast, our layer-by-layer (LbL) self-assembly approach is both effective and scalable, making it cost-efficient for industrial applications. To evaluate the performance under high current density and mass loading, we tested both Zn@CMC and Zn@PAH/PAA electrodes in a 15Ah full cell to evaluate actual anode behavior. As illustrated in Figure S20 below, Zn@PAH/PAA maintained stable GCD performance above 15 Ah with nearly 100% CE value over 200 cycles at a charging rate of nearly 2C. In contrast, Zn@CMC exhibited a drop in CE around 85 cycles, followed by a rapid capacity loss. The inset shows post-analysis revealing delamination between the CMC coating and Zn anode. This delamination led to wrinkling in both the separator and cathode, causing a sharp decline in CE. Since side reactions were not isolated, the monolayer coating easily delaminated. However, the PAH/PAA multilayers can effectively separate side reactions, allowing for a stable and robust plating regime, which in turn improved battery life under high charging rates.

Figure S20. (a) Pouch cell of Zn@PAH/PAA in 15Ah; (b) Pouch cell of Zn@CMC in 15Ah (Anode coating layer delamination).

The above additional tests clearly illustrate the superior performance of the PAH/PAA multilayers coated Zn anode compared to the CMC monolayer coated Zn anode, particularly in terms of maintaining structural integrity and CE value over extended cycling. One of the most critical benefits of the PAH/PAA multilayers is its ability to prevent delamination. Secondly, the PAH/PAA coating also excels at separating side reactions, which are a major cause of anode surface instability in ZIBs. By isolating these reactions, the PAH/PAA multilayers facilitate to create a uniform and stable plating/stripping process on the Zn anode. This ensures a smooth electrode surface, which minimizes the formation of dendrites and reduces passivation, both of which are detrimental to battery performance. This is especially critical for large-scale and high-performance applications where high current densities are required. The stable performance of the PAH/PAA multilayers at high rates shows that it can handle the demands of industrial applications without compromising efficiency or lifespan. We have further expanded on these points in the revised manuscript.

Revised manuscript Page 1

The dual-ion channels, created by strong electrostatic interactions between carboxylate anions (COO^-) and ammonia cations (NH_3^+), block SO_4^{2-} and promote the uniform Zn deposition along the (002) plane, exhibiting a CE of 99.8% after 1600 cycles in the Zn||Cu asymmetric cell. With the facile fabrication of the layer-by-layer self-assembled Zn anode, an Ah-level pouch cell (17.36 Ah) with a high mass loading ($> 8 \text{ mg cm}^{-2}$) demonstrated the practical viability for large-scale applications, retaining a capacity of 93.6% for at least 250 cycles at 1.7 C. This innovation not only enables faster and more uniform Zn deposition but also significantly

enhances the cycling stability and mechanical durability in larger pouch cells, paving the way for the commercialization of AZIBs.

Page 3

Based on this, the PAH/PAA multilayers could be seen as the dual-ion channels for SO_4^{2-} and Zn^{2+} in the electrolyte, like an ionic separation, the dual-ion channels block SO_4^{2-} and attract Zn^{2+} , which would regulate the mobility and dispersion of Zn^{2+} and suppress the side reactions. Moreover, the strong electrostatic interactions between PAH and PAA could effectively improve the mechanical strength without affecting the ionic conductivity of the coating, and also simplify the preparation process.^{19,27} As illustrated in Scheme 1b, the preparation sequence of multilayers is to first coat the PAH layer and then the PAA layer (Anode⁻ – PAH⁺ – PAA⁻), followed by a rinsing process after each coating to remove the weakly associated bound chains. Designing in the sequence: Anode⁻ – PAH⁺ – PAA⁻ would enable a high adhesion to the negatively charged Zn anode and increase the zincophilicity of multilayers. PAA layer as the outer layer could first regulate the diffusion of Zn^{2+} and repel SO_4^{2-} to a certain degree, while PAH layer would further capture SO_4^{2-} due to the low binding energy. Indeed, the PAH/PAA multilayers lead to the formation of ion-separation accelerating channels to block SO_4^{2-} and accelerate Zn^{2+} transference, thereby promoting the uniform Zn deposition and inhibiting the HER and by-products.

Page 10

To assess performance under high current density and mass loading, both Zn@CMC and Zn@PAH/PAA electrodes in a 15Ah full cell were analyzed to evaluate actual anode behavior. As illustrated in Figure S20, Zn@PAH/PAA maintains stable GCD performance above 15 Ah with nearly 100% CE value over 200 cycles at a charging rate of nearly 2 C. In contrast, Zn@CMC exhibits a drop around 85 cycles, followed by a rapid capacity loss. The inset shows post-analysis revealing delamination between the CMC coating and the Zn anode. This delamination leads to wrinkling in both the separator and cathode, causing a sharp decline in CE value. Since side reactions are not well suppressed, the monolayer coating easily delaminates. In contrast, the PAH/PAA multilayers effectively separate side reactions, allowing for a stable and robust plating regime, which in turn improved battery life under high charging rates. These results illustrate the superior performance of the PAH/PAA multilayers compared to the mono hydrophilic polyelectrolyte coatings, particularly in terms of maintaining structural integrity and CE value over extended cycling.

2. And I could not agree that this assembly can enable an ion-sieving channel, which may exaggerate. Two different charged polymers will naturally attract oppositely charged ions. However, it does not imply an ion channel, which is often achieved by metal-organic frameworks or covalent-organic frameworks.

Response: Thank you for your follow-up comment. I understand the concern regarding the terminology "ion-sieving channel," and I appreciate the comparison with more structured frameworks like metal-organic frameworks (MOFs) or covalent-organic frameworks (COFs). However, I'd like to clarify that the term "ion-sieving" in our work is not intended to imply a

rigid and well-defined channel as in MOFs or COFs but rather a dynamic, electrostatically driven process facilitated by the PAA/PAH multilayers. The formation of these dual-ion channels is achieved through layer-by-layer (LbL) assembly, where electrostatic interactions between the oppositely charged PAA (negatively charged) and PAH (positively charged) provide selective pathways for Zn^{2+} and SO_4^{2-} . While MOFs and COFs typically rely on their structural porosity for ion sieving, our system exploits the electrostatic potential across the multilayer interface to regulate ion diffusion. To further understand the mechanism on ion-separation, we have conducted XPS for verification. S 2p spectra for the cycled Zn@PAA/PAH electrode was characterized by X-ray photoelectron spectroscopy (XPS), as illustrated in Figure S6. The detected peaks at approximately 168 eV and 162 eV correspond to SO_4^{2-} and ZnS contents, respectively.^{1,2} After Ar^+ etching, the signal of SO_4^{2-} becomes faint, while the signal of ZnS markedly increases. These results indicate that the PAH/PAA multilayers could significantly block SO_4^{2-} and suppress the HER and side reactions to enhance the thermostability of Zn anodes. This result also validates the XRD pattern as shown in last comment reply. Moreover, the *in-situ* Raman spectra also convince the function of the ion regulating dual-ion channels. These regular and reversible band changes indicate the formation of dual-ion channels between PAH and PAA polyelectrolytes. The binding energy was calculated to further discuss the interfacial mechanism. As shown in Figure 4b, the binding energy of $[\text{PAA}^- \text{Zn}(\text{H}_2\text{O})_4]^+$ decreases from -12.913 eV to -15.700 eV, combined with the activation energy calculation (Figure 1c), which indicates that PAA^- would coordinate with Zn^{2+} to form the solvation structure of $[\text{PAA}^- \text{Zn}(\text{H}_2\text{O})_4]^+$, thereby regulating the Zn^{2+} diffusion.

Figure S6. XPS spectra of S 2p for the cycled Zn@PAH/PAA electrode before and after Ar^+ etching (etching depth was set to 50 nm (TiO_2 reference)). (Cycling parameter: 50 cycles at 0.5 mA cm^{-2} and 0.5 mAh cm^{-2}).

Figure S16. *In-situ* Raman spectra of different substrates.

Moreover, recent studies on LbL films, particularly those using polyelectrolytes like PAA and PAH, show that they can form adaptive and flexible nanostructures that allow selective ion transport. For instance, Cao et al. demonstrated that surface patterns and ion interactions can significantly alter the transport properties of LbL films under specific conditions.^{3,4} This flexibility and adaptability are key in our system, where we focus on preventing the buildup of harmful by-products like $\text{Zn}_4\text{SO}_4(\text{OH})_6 \cdot x\text{H}_2\text{O}$ through the selective regulation of SO_4^{2-} ions, even though it may not form a traditional ion channel as seen in MOFs. In essence, our PAA/PAH system should be viewed as a dynamic ion-separation interface rather than a static channel, with the primary advantage being its tunable, scalable nature for practical applications. This electrostatic control mechanism is effective for ion regulation and enhances the performance and stability of the Zn anode, which is evident in the improved cycling life and reduced formation of corrosion by-products.

To further clarify this novelty, we changed the terminology to “ion-separation” instead. Layer-by-layer deposition of polycations and polyanions multilayers on the surface of anion exchange membranes (AEMs) is a simple and versatile method to obtain monovalent anion selectivity. However, the stability of the polyelectrolyte multilayers (PEMs) can be compromised by the weak interactions formed between the deposited barrier and the pristine membrane surface. In this sense, cross-linking appears as an efficient method to improve the chemical stability of PEMs by covalent bonding.

Revised manuscript Page 4

To further understand the mechanism on suppressing by-products, S 2p spectra for the cycled Zn@PAA/PAH electrode was characterized by X-ray photoelectron spectroscopy (XPS), as illustrated in Figure S6. The detected peaks at approximately 168 eV and 162 eV correspond to SO_4^{2-} and ZnS contents, respectively.^{35,36} After Ar^+ etching, the signal of SO_4^{2-} becomes faint, while the signal of ZnS markedly increases. These results indicate that the PAH/PAA multilayers could significantly block SO_4^{2-} and suppress the HER and side reactions to enhance

the thermostability of Zn anodes. To evaluate the ability of the LbL self-assembled polyelectrolytes coating to suppress side reactions compared to other hydrophilic polyelectrolyte coatings, a series of comparative experiments with the sodium alginate (SA) and carboxymethyl cellulose (CMC) coatings were conducted. As shown in the Figure S7, although there is no huge difference regarding the plating overpotential, Zn@PAH/PAA still exhibits the relatively lowest magnitude compared to other monolayer hydrophilic polyelectrolyte SA and CMC. Moreover, XRD patterns reveal that compared to the SA coating, the PAH/PAA multilayers can more effectively suppress side reactions, as merely $\text{Zn}_4\text{SO}_4(\text{OH})_6 \cdot 4\text{H}_2\text{O}$ byproduct was detected. This result illustrates that although monolayer polyelectrolyte coatings can promote homogeneous Zn deposition by regulating Zn^{2+} flux, their ability to repel SO_4^{2-} ions are limited, especially under high current densities. In contrast, the ion-separation accelerating channels constructed by LbL self-assembled polyelectrolytes can better inhibit side reactions.

3. Nevertheless, this is good work showing incremental advances in optimizing the utilization of polymer electrolytes for zinc-ion batteries and demonstrating the pouch battery application, but not enough novelty and general interest for *Nature Communication*. The work is more suitable for an energy-related journal.

Response: Thank you for your constructive feedback. We acknowledge that optimizing polyelectrolytes for ZIBs is a well-explored area, but our work presents a significant advance in the practical scalability of ZIBs technology, which is crucial for real-world applications. The novelty of our work lies not only in the material innovation but also in solving critical issues related to the scaling up of ZIBs for industrial and commercial use—an area that has seen limited advancement. One of the persistent challenges in ZIBs development is maintaining long-term stability and high performance under high current densities and large cell sizes, conditions that conventional coating strategies fail to address. While traditional methods perform well in coin cell tests, they are often far from practical in larger-scale applications, where side reactions such as dendrite formation and anode passivation become significantly more problematic. Our layer-by-layer (LbL) self-assembly approach, with its dual-ion channel system, is both novel and scalable, allowing for precise control over the diffusion of Zn^{2+} and SO_4^{2-} , which helps to minimize side reactions, even in large pouch cells. This design not only enhances the mechanical strength, adhesion, and long-term durability of the anode but also offers a cost-efficient solution that can be easily integrated into current manufacturing processes. Importantly, our method bridges the gap between the laboratory-scale innovation and the real-world industrial application, which remains a significant barrier in the advancement of ZIBs technology.

We believe the scalability, industrial relevance, and emphasis on practical application set this work apart from incremental advances, making it highly relevant to a broader scientific and industrial community. These features align well with the impact and scope of *Nature Communications*, contributing to the ongoing efforts to make commercially viable ZIBs to reality.

Ref:

- [1]. Yang, J. *et al.* Three Birds with One Stone: Tetramethylurea as Electrolyte Additive for Highly Reversible Zn-Metal Anode. *Advanced Functional Materials* **32**, 2209642 (2022).
- [2]. Wang, Y. *et al.* Suppressed Water Reactivity by Zincophilic-Hydrophobic Electrolyte Additive for Superior Aqueous Zn Metal Batteries. *Advanced Energy Materials* **13**, 2302707 (2023).
- [3]. Cao, M., Wang, J. & Wang, Y. Surface patterns induced by Cu²⁺ ions on BPEI/PAA layer-by-layer assembly. *Langmuir* **23**, 3142-3149 (2007).
- [4]. Peng, N. *et al.* Fabrication and stability of porous poly (allylamine) hydrochloride (PAH)/poly (acrylic acid)(PAA) multilayered films via a cleavable-polycation template. *Polymer* **52**, 1256-1262 (2011).

Reviewer 2

In this work, the authors report a method to form polyelectrolyte layer to promote uniform Zn diffusion and deposition on the anode for Zinc-ion batteries. The concept of the polyelectrolyte layer formation via LbL is not new, but the authors make an interesting claim that the layer acts as an ion-sieving accelerating channel, which remains to be further validated. The reported performance of the ZIB is impressive and the authors clearly know the current challenges in the subject. However, the PAH/PAA chemistry has been around for a long time, and the LbL method is also a well-established technique. Although this is an interesting work with good results, in the Reviewer's opinion, this manuscript does not seem to provide "new" chemistry nor innovative method to be considered in the Nature communication journal.

Response: Thank you for your detailed feedback and for acknowledging the significance of our results. We appreciate your recognition of the challenges in AZIBs development and agree that polyelectrolyte chemistry and the layer-by-layer (LbL) method have been well-established techniques. However, the novelty of our work lies not in introducing new chemistry, but rather in how we have re-engineered these known materials and techniques to address a key bottleneck in ZIBs performance: the mitigation of side reactions and non-uniform Zn deposition, particularly at larger scales and higher current densities.

Our claim of the ion-separation accelerating channel is based on the innovative use of the PAH/PAA multilayers to create dual-ion channels that can regulate the diffusion of both Zn^{2+} and SO_4^{2-} . While the PAH/PAA chemistry system and the LbL self-assembly method are not new, their application in such a dual-ion regulation mechanism, especially in the context of preventing the formation of detrimental by-products like $Zn_4SO_4(OH)_6 \cdot xH_2O$ under industrial-scale conditions, is a new and practical innovation. This approach directly addresses the challenges faced by traditional designs of AZIBs, which often struggle with performance degradation under high current densities and in larger cell formats.

Our focus is on making ZIBs technology scalable for real-world applications, an area where much of the previous literature falls short. While the LbL technique has been explored in other applications, its use in developing a cost-effective, scalable situation for the Zn anode modification represents a significant innovation. The synergy between material chemistry and practical scalability in our work not only provides a new insight for the surface modification of the Zn anode but also offers a feasible pathway towards the commercialization of AZIBs. We believe that the combination of well-established techniques and innovative strategies can address both fundamental scientific challenges and industrial-scale applications, thereby promoting the development of AZIBs. We hope that these aspects can demonstrate the novelty and broader relevance of our work. Thank you again for your constructive comments.

1. Can authors please provide direct proof that the PAH/PAA layer acts as an ion-sieving "accelerating" channel? How does it accelerate the Zn ion flux? It has been well-reported that simply making the surface hydrophilic improves the performance.

Response: Thank you for your insightful comment. We appreciate your request for more direct

evidence of the PAH/PAA multilayers functioning as an ion-separation “accelerating” channel. To substantiate our claim, we performed a series of experimental analyses to demonstrate how the PAH/PAA multilayers enhance the Zn^{2+} ion flux and go beyond the effects of only increasing surface hydrophilicity.

While increasing the surface hydrophilicity can improve the Zn^{2+} ion flux, it is insufficient to rely solely on this under high current densities. Moreover, the mechanical strength and capacity to regulate Zn^{2+} and SO_4^{2-} diffusion offered by a simple hydrophilic interface are limited. Instead, the ion-separation mechanism constructed by the PAH/PAA multilayers we proposed, specifically its ability to regulate both Zn^{2+} and SO_4^{2-} flux, plays an important role in accelerating Zn^{2+} transport and blocking SO_4^{2-} to inhibit by-products.

The formation of ion-separation “accelerating” channels is achieved through layer-by-layer (LbL) assembly, where electrostatic interactions between the oppositely charged PAA (negatively charged) and PAH (positively charged) provide selective pathways for Zn^{2+} and SO_4^{2-} ions. Moreover, the preparation sequence of multilayers is to first coat the PAH layer and then the PAA layer (Anode⁻ – PAH⁺ – PAA⁻). PAA layer as the outer layer could first regulate the diffusion of Zn^{2+} and repel SO_4^{2-} to a certain degree, while PAH layer would further capture SO_4^{2-} due to the low binding energy. Indeed, the PAH/PAA multilayers lead to the formation of ion-separation “accelerating” channels to block SO_4^{2-} and accelerate Zn^{2+} transference.

To prove the mechanism on ion-separation, we have conducted XPS, XRD, and *in-situ* Raman for verification. S 2p spectra for the cycled Zn@PAA/PAH electrode was characterized by X-ray photoelectron spectroscopy (XPS), as illustrated in Figure S6. The detected peaks at approximately 168 eV and 162 eV correspond to SO_4^{2-} and ZnS contents, respectively.^{1,2} After Ar⁺ etching, the signal of SO_4^{2-} becomes faint, while the signal of ZnS markedly increases. XRD patterns of the Bare Zn electrode after 50 cycles exhibits strong diffraction peaks of the by-product ($\text{Zn}_4\text{SO}_4(\text{OH})_6 \cdot 4\text{H}_2\text{O}$, ZHS), which was not detected on the cycled Zn@PAH/PAA electrode (Figure 1b). These results indicate that the PAH/PAA multilayers could significantly block SO_4^{2-} and suppress the HER and side reactions to enhance the thermostability of Zn anodes. Moreover, the *in situ* Raman spectra also convince the function of the ion-separation channels. These regular and reversible band changes indicate the formation of dual-ion channels between PAH and PAA polyelectrolytes. The Zn@PAH/PAA anode maintains a CE value of 99.8% over 1,600 cycles, compared to the Bare Zn anode, which exhibits significant capacity degradation after 70 cycles. This stability is largely due to the PAH/PAA multilayers’ ion-separation capabilities, which promote uniform Zn deposition, prevent dendrite formation, and reduce the impact of parasitic side reactions.

Figure S6. (a) XPS spectra of S 2p for the cycled Zn@PAH/PAA electrode before and after Ar⁺ etching (etching depth was set to 50 nm (TiO₂ reference)). (Cycling parameter: 50 cycles at 0.5 mA cm⁻² and 0.5 mAh cm⁻²).

Figure S16. *In-situ* Raman spectra of different substrates.

To directly examine the ion transport properties of the PAH/PAA multilayers, we conducted the chronoamperometry (CA) test and calculated the Zn transference number, both of which provide insights into ion diffusion and charge transfer dynamics. As shown in Figure 1f, the CA test reflects that Zn²⁺ exhibits a 2D diffusion behavior on Zn@PAH/PAA, compared with a 3D diffusion on the Bare Zn. This suggests uniform deposition on the Zn@PAH/PAA electrode, but aggregation occurs on the Bare Zn electrode. Moreover, the current for Zn deposition was consistently higher for the Zn@PAH/PAA electrode compared to the Bare Zn electrode, indicating the accelerated Zn²⁺ ion flux. Zn transference number of Zn@PAH/PAA and Bare Zn were calculated in Figure 1d and Figure S9, where the high ionic conductivity of PAH/PAA multilayers can increase the Zn transference number from 0.284 to 0.481. These results suggest the ion-separation channels can modulate and accelerate interfacial kinetics of Zn²⁺ diffusion. The binding energy was calculated to further discuss the interfacial transference mechanism. As shown in Figure 4b, the binding energy of [PAA⁻–Zn(H₂O)₄]⁺ decreases from -12.913 eV to -15.700 eV, combined with the activation energy calculation (Figure 1c), which indicates that PAA⁻ would coordinate with Zn²⁺ to form the solvation structure of [PAA⁻–Zn(H₂O)₄]⁺, thereby

regulating the Zn^{2+} diffusion.

Results from our electrochemical analyses and DFT calculations provide compelling evidence that the PAH/PAA multilayers act as ion-separation “accelerating” channels. The multilayers cannot only regulate the solvation structure of Zn^{2+} and reduce the charge transfer resistance to accelerate the ion flux, but also block SO_4^{2-} to suppress the formation of by-products, offering a robust solution for improving ZIBs performance under industrially relevant conditions beyond the effects of a mono hydrophilic interface. We hope these results address your concern and offer clear validation of the proposed ion-separation mechanism. Thank you for your thoughtful feedback. We have further expanded on these points in the revised manuscript.

Revised manuscript Page 1

The dual-ion channels, created by strong electrostatic interactions between carboxylate anions (COO^-) and ammonia cations (NH_3^+), block SO_4^{2-} and promote the uniform Zn deposition along the (002) plane, exhibiting a CE of 99.8% after 1600 cycles in the Zn||Cu asymmetric cell. With the facile fabrication of the layer-by-layer self-assembled Zn anode, an Ah-level pouch cell (17.36 Ah) with a high mass loading ($> 8 \text{ mg cm}^{-2}$) demonstrated the practical viability for large-scale applications, retaining a capacity of 93.6% for at least 250 cycles at 1.7 C. This innovation not only enables faster and more uniform Zn deposition but also significantly enhances the cycling stability and mechanical durability in larger pouch cells, paving the way for the commercialization of AZIBs.

Page 3

Based on this, the PAH/PAA multilayers could be seen as the dual-ion channels for SO_4^{2-} and Zn^{2+} in the electrolyte, like an ionic separation, the dual-ion channels block SO_4^{2-} and attract Zn^{2+} , which would regulate the mobility and dispersion of Zn^{2+} and suppress the side reactions. Moreover, the strong electrostatic interactions between PAH and PAA could effectively improve the mechanical strength without affecting the ionic conductivity of the coating, and also simplify the preparation process.^{19,27} As illustrated in Scheme 1b, the preparation sequence of multilayers is to first coat the PAH layer and then the PAA layer (Anode⁻ – PAH⁺ – PAA⁻), followed by a rinsing process after each coating to remove the weakly associated bound chains. Designing in the sequence: Anode⁻ – PAH⁺ – PAA⁻ would enable a high adhesion to the negatively charged Zn anode and increase the zincophilicity of multilayers. PAA layer as the outer layer could first regulate the diffusion of Zn^{2+} and repel SO_4^{2-} to a certain degree, while PAH layer would further capture SO_4^{2-} due to the low binding energy. Indeed, the PAH/PAA multilayers lead to the formation of ion-separation accelerating channels to block SO_4^{2-} and accelerate Zn^{2+} transference, thereby promoting the uniform Zn deposition and inhibiting the HER and by-products.

Page 4

Revised manuscript Page 4

To further understand the mechanism on suppressing by-products, S 2p spectra for the cycled

Zn@PAA/PAH electrode was characterized by X-ray photoelectron spectroscopy (XPS), as illustrated in Figure S6. The detected peaks at approximately 168 eV and 162 eV correspond to SO_4^{2-} and ZnS contents, respectively.^{35,36} After Ar^+ etching, the signal of SO_4^{2-} becomes faint, while the signal of ZnS markedly increases. These results indicate that the PAH/PAA multilayers could significantly block SO_4^{2-} and suppress the HER and side reactions to enhance the thermostability of Zn anodes. To evaluate the ability of the LbL self-assembled polyelectrolytes coating to suppress side reactions compared to other hydrophilic polyelectrolyte coatings, a series of comparative experiments with the sodium alginate (SA) and carboxymethyl cellulose (CMC) coatings were conducted. As shown in the Figure S7, although there is no huge difference regarding the plating overpotential, Zn@PAH/PAA still exhibits the relatively lowest magnitude compared to other monolayer hydrophilic polyelectrolyte SA and CMC. Moreover, XRD patterns reveal that compared to the SA coating, the PAH/PAA multilayers can more effectively suppress side reactions, as merely $\text{Zn}_4\text{SO}_4(\text{OH})_6 \cdot 4\text{H}_2\text{O}$ byproduct was detected. This result illustrates that although mono polyelectrolyte coatings can promote homogeneous Zn deposition by regulating Zn^{2+} flux, their ability to repel SO_4^{2-} ions are limited, especially under high current densities. In contrast, the ion-separation accelerating channels constructed by LbL self-assembled polyelectrolytes can better inhibit side reactions.

2. The change in the interfacial activation energy shown in this work could have been obtained by other hydrophilic modification. What is different and unique about PAH/PAA compared to other hydrophilic modification works? The authors should provide a convincing control experiment where another hydrophilic modification is used with similar surface properties.

Response: Thank you for raising this important point. To address your comment and highlight the uniqueness of the PAH/PAA multilayers compared to other hydrophilic modifications, we have conducted comparative experiments with sodium alginate (SA) and carboxymethyl cellulose (CMC) hydrophilic coatings to evaluate the difference and unique about the PAH/PAA multilayers in performance, particularly in terms of the inhibition of by-products and the large-scale applications.

Although there have been several studies on using the hydrophilic modification to improve the performance of AZIBs, most of these works focus on using a single polyelectrolyte to protect the anode interface. However, as mentioned above, a single hydrophilic polyelectrolyte cannot provide sufficient mechanical strength and capacity to inhibit the formation of by-products, especially under high current conditions. Our previous work utilized the polyanionic electrolyte sodium alginate (SA) to enhance the performance of the Zn anode.¹ While this approach showed promise, we observed certain limitations. Specifically, a mono-polyelectrolyte interface has restricted control over ion diffusion and provides limited mechanical strength to the anode surface. Based on this, we propose the LbL self-assembly technique to construct dual-ion channels, which can effectively regulate the diffusion of both SO_4^{2-} and Zn^{2+} and enhance mechanical strength. Moreover, benefiting from the industrial maturity of the LbL self-assembly technology, we scaled up the battery performance tests to large pouch cells. Our primary objective is to make ZIBs scalable for practical applications, a gap that much of the

current research has not fully addressed. We are confident that combining the established method with the novel approach can tackle both core scientific challenges and large-scale applications, thereby advancing the development of AZIBs.

Thank you for highlighting the interfacial activation energy presented in our work. Although the change in the activation energy shown in our work could have been obtained by other hydrophilic modifications, it does not fully reflect the overall performance of batteries. Through the activation energy test, we simply aimed to verify that the PAH/PAA multilayers can efficiently regulate the solvation structure of Zn^{2+} . The novelty and distinction of our proposed self-assembled polyelectrolytes strategy lie in its ability to better suppress side reactions and facilitate the large-scale applications of AZIBs.

We have conducted additional tests to clarify the uniqueness and difference. As shown in Figure S7 below, although there is no huge difference regarding the plating overpotential, Zn@PAH/PAA still exhibits the relatively lowest magnitude compared to other monolayer hydrophilic polyelectrolyte SA and CMC. Although anionic polyelectrolytes can promote homogeneous Zn deposition by regulating Zn^{2+} flux, they show a limited ability to repel SO_4^{2-} , particularly under high current densities. This can lead to the formation of by-products. By comparing the XRD patterns between cycled Zn@PAH/PAA and Zn@SA, we can clearly see that the PAH/PAA multilayers can more effectively suppress side reactions, as merely $Zn_4SO_4(OH)_6 \cdot 4H_2O$ byproduct was detected. Our approach creates dual channels that simultaneously regulate the diffusion of both Zn^{2+} and SO_4^{2-} ions. The strong electrostatic interactions between the carboxylate anions (COO^-) in PAA and the ammonium cations (NH_3^+) in PAH create a separation mechanism that block SO_4^{2-} , preventing their participation in undesirable side reactions. This capture mitigates the formation of insulating layers of by-products, which are a common challenge in conventional systems using mono-polyelectrolytes. As shown in Figure 3c, the Zn||Zn performance is still stable under a high DoD of 53.4%. The nucleation radius for Zn@PAH/PAA and Bare Zn is 0.47, which is attributed to an increased nucleation rate and a uniform plating.

Figure S7. (a) Optical image after coating in mono and multilayers. (b) Zn||Zn performance. (c) XRD comparisons after 50 cycles at 1 mA cm⁻² and 1 mAh cm⁻².

In addition, while traditional coating methods have shown good results in coin cell tests, they are often not feasible for large-scale applications. In contrast, our layer-by-layer (LbL) self-assembly approach is both effective and scalable, making it cost-efficient for industrial use. Due to its ability to inhibit by-products, the Zn@PAH/PAA electrode maintains a high Coulombic efficiency of 99.8% over 1,600 cycles, significantly reducing passivation and dendrite formation compared to conventional systems. This results in a longer lifespan and greater practical viability for ZIBs. To assess performance under high current density and mass loading, we tested both Zn@CMC and Zn@PAH/PAA coatings in a 15Ah full cell to evaluate actual anode behavior. As illustrated in Figure S20, Zn@PAH/PAA maintains stable GCD performance above 15 Ah with nearly 100% Coulombic efficiency over 200 cycles at a charging rate of nearly 2 C. In contrast, Zn@CMC exhibits a drop in efficiency around 85 cycles, followed by a rapid capacity loss. The inset shows post-analysis revealing delamination between the CMC coating and the Zn anode. This delamination leads to wrinkling in both the separator and cathode, causing a sharp decline in CE. Since side reactions are not well suppressed, the monolayer coating easily delaminates. In contrast, the PAH/PAA multilayers effectively separate side reactions, allowing for a stable and robust plating regime, which in turn improved battery life under high charging rates.

Figure S20. (a) Pouch cell of Zn@PAH/PAA in 15Ah; (b) Pouch cell of Zn@CMC in 15Ah (Anode coating layer delamination).

The above additional tests clearly illustrate the superior performance of the PAH/PAA multilayers compared to the single hydrophilic polyelectrolyte, particularly in terms of maintaining structural integrity and CE value over extended cycling. One of the most critical benefits of the PAH/PAA multilayers is its ability to prevent delamination. Secondly, the PAH/PAA coating is more effective at inhibiting side reactions. By isolating these reactions, the PAH/PAA multilayer help to create a uniform and stable plating/stripping process on the Zn anode. This ensures a smooth electrode surface, which minimizes the formation of dendrites and

reduces passivation, both of which are detrimental to battery performance. This is especially critical for large-scale and high-performance applications where high current densities are required. The stable performance of the PAH/PAA multilayers at high rates shows that it can handle the demands of industrial applications without compromising efficiency or lifespan. We hope this additional data convincingly addresses your concern. Thank you again for your valuable feedback.

Revised manuscript Page 3

Based on this, the PAH/PAA multilayers could be seen as the dual-ion channels for SO_4^{2-} and Zn^{2+} in the electrolyte, like an ionic separation, the dual-ion channels block SO_4^{2-} and attract Zn^{2+} , which would regulate the mobility and dispersion of Zn^{2+} and suppress the side reactions. Moreover, the strong electrostatic interactions between PAH and PAA could effectively improve the mechanical strength without affecting the ionic conductivity of the coating, and also simplify the preparation process.^{19,27} As illustrated in Scheme 1b, the preparation sequence of multilayers is to first coat the PAH layer and then the PAA layer (Anode⁻ – PAH⁺ – PAA⁻), followed by a rinsing process after each coating to remove the weakly associated bound chains. Designing in the sequence: Anode⁻ – PAH⁺ – PAA⁻ would enable a high adhesion to the negatively charged Zn anode and increase the zincophilicity of multilayers. PAA layer as the outer layer could first regulate the diffusion of Zn^{2+} and repel SO_4^{2-} to a certain degree, while PAH layer would further capture SO_4^{2-} due to the low binding energy. Indeed, the PAH/PAA multilayers lead to the formation of ion-separation accelerating channels to block SO_4^{2-} and accelerate Zn^{2+} transference, thereby promoting the uniform Zn deposition and inhibiting the HER and by-products.

Page 10

To assess performance under high current density and mass loading, both Zn@CMC and Zn@PAH/PAA electrodes in a 15Ah full cell were analyzed to evaluate actual anode behavior. As illustrated in Figure S20, Zn@PAH/PAA maintains stable GCD performance above 15 Ah with nearly 100% CE value over 200 cycles at a charging rate of nearly 2 C. In contrast, Zn@CMC exhibits a drop around 85 cycles, followed by a rapid capacity loss. The inset shows post-analysis revealing delamination between the CMC coating and the Zn anode. This delamination leads to wrinkling in both the separator and cathode, causing a sharp decline in CE value. Since side reactions are not well suppressed, the monolayer coating easily delaminates. In contrast, the PAH/PAA multilayers effectively separate side reactions, allowing for a stable and robust plating regime, which in turn improved battery life under high charging rates. These results illustrate the superior performance of the PAH/PAA multilayers compared to the mono hydrophilic polyelectrolyte coatings, particularly in terms of maintaining structural integrity and CE value over extended cycling.

3. The preferential nucleation of Zn^{2+} along Zn(002) surface is very interesting. Has this been observed before? Do the authors expect the same outcome on different hydrophilic modification methods? Again, similarly to my previous comment, what is unique about PAH/PAA chemistry that leads to preferential nucleation? I think the DFT works should be moved into the main

manuscript, if it can provide convincing evidence.

Response: Thank you for your interest in the results demonstrating that the PAH/PAA multilayers can induce the preferential nucleation of Zn^{2+} along the Zn(002) plane. The PAH/PAA multilayers were prepared in the sequence of Anode⁻ – PAH⁺ – PAA⁻, forming the dual-ion channels that block SO_4^{2-} and attract Zn^{2+} in the structure of Anode⁻ – PAH⁺ – SO_4^{2-} – $\text{Zn}(\text{H}_2\text{O})_4^{2+}$ – PAA⁻. According to SEM images, TEM images, XRD patterns, and DFT calculations, we propose that such charged ionic channels, constructed by the PAH and PAA polyelectrolytes, can induce a preferential deposition along Zn(002) plane.

There have been some research about the Zn transference interface constructed by hydrophilic ionic polyelectrolytes can induce the oriented deposition along (002) plane. Jiao and Wu et al. reported a zwitterionic gel poly(3-(1-vinyl-3-imidazolio) propanesulfonate), where the charged group in the gel electrolyte can texture Zn^{2+} nucleation and deposition to (002), thereby inducing uniform deposition.³ Xiang et al. developed a self-adaptable polydimethylsiloxane (PDMS)/montmorillonite (MMT) film, in which the adsorption of oxygen-containing sited on PDMS and Zn^{2+} nanochannels in the negative charged MMT interlayer can capture Zn^{2+} and induce horizontally oriented Zn(002) deposition.⁴ Our previous work, as mentioned above, utilized the polyanionic electrolyte sodium alginate (SA) to construct an “egg-box” shaped accelerating channel for Zn^{2+} , which also can promote Zn(002) deposition.¹

However, although such hydrophilic ionic polyelectrolytes can promote Zn(002) deposition, their ability to inhibit side reactions is limited (as illustrated above: XRD patterns of cycled Zn@PAH/PAA and Zn@SA), and the mechanism for regulating SO_4^{2-} diffusion has not been addressed. In this work, we utilized XRD, XPS, *in situ* Raman, and DFT calculations to figure out the mechanism of the constructed ion-separation channels for regulating both Zn^{2+} and SO_4^{2-} . Moreover, the novelty of our work lies in demonstrating that the PAH/PAA multilayers not only induce Zn(002) deposition and inhibit side reactions, but also that the LbL self-assembled polyelectrolytes strategy can be effectively applied to large-scale applications. In small coin cells, regulating Zn nucleation and deposition has been relatively manageable, but scaling up to larger pouch cells poses a significant challenge due to increased current densities and uneven Zn^{2+} distribution across the whole electrode surface. As mentioned in Figure S20 above, we tested both Zn@CMC and Zn@PAH/PAA coatings in a 15Ah full cell to evaluate actual anode behavior. As there is no strong interaction between Zn anode with the CMC layer, it is easy to get delamination over a long cycling and under high current. This innovation enables not only faster and more uniform Zn deposition but also greatly improved cycling stability and mechanical durability in larger pouch cells, paving the way for commercially viable ZIBs technology.

We agree with your suggestion to move the DFT calculations into the main manuscript to provide a more robust explanation of the unique behavior of the PAH/PAA multilayers as shown in Figure 4c. DFT simulation results reveal that the Zn^{2+} adsorption energy on the Zn(002) surface with the PAH/PAA multilayers is significantly reduced, confirming that the electrostatic interactions and the surface energy modulation induced by the polyelectrolyte layers favor

Zn(002) nucleation. These calculations provide convincing evidence that the PAH/PAA multilayers promotes more stable and uniform Zn²⁺ nucleation compared to other hydrophilic coatings. The LbL self-assembled polyelectrolytes strategy provides a unique combination of the ion-separation capability, electrostatic control, and a layered structure that actively promotes preferential Zn(002) nucleation, an advantage that cannot be replicated by other hydrophilic modifications. Thanks for your suggestion, we have removed the DFT data to the main manuscript to further strengthen the evidence for this claim.

To further clarify this novelty, we changed the terminology to “ion-separation” instead. Layer-by-layer deposition of polycations and polyanions multilayers on the surface of anion exchange membranes (AEMs) is a simple and versatile method to obtain monovalent anion selectivity. However, the stability of the polyelectrolyte multilayers (PEMs) can be compromised by the weak interactions formed between the deposited barrier and the pristine membrane surface. In this sense, cross-linking appears as an efficient method to improve the chemical stability of PEMs by covalent bonding.

Revised manuscript Page 9

The interaction of SO₄²⁻ and Zn@PAH/PAA was also investigated (Figure 4c and Table S4). The binding energy of [Zn(SO₄)₂]²⁻ is much larger than that of ZnSO₄ (-10.068 eV and -3.785 eV, respectively), suggesting that Zn²⁺ is likely to bind with two SO₄²⁻ to form [Zn(SO₄)₂]²⁻. In addition, compared with [(PAH⁺)₃ SO₄]⁺ and [(PAA⁻)₁ SO₄]³⁻, [(PAH⁺)₃ Zn(SO₄)₂]⁺ exhibits the lowest binding energy (-26.33 eV), indicating that SO₄²⁻ would bind with PAH⁺ to form the stable [(PAH⁺)₃ Zn(SO₄)₂]⁺ coordination structure. Based on the above results, the interfacial mechanism during Zn²⁺ plating/stripping on the Zn@PAH/PAA electrode is illustrated in Figure 4d. The ion-separation accelerating channels in the structure of PAH⁺ – SO₄²⁻ – Zn(H₂O)₄²⁺ – PAA⁻ is constructed by the LbL self-assembled PAH/PAA multilayers, where PAA⁻ would regulate the Zn²⁺ solvation shell and accurate Zn²⁺ transport at the inner Helmholtz plane, and PAH⁺ would bind with SO₄²⁻ to inhibit the formation of ZHS. Indeed, the PAH/PAA multilayers would also induce the Zn nucleation and deposition along (002) texture to form the uniform and dense Zn flake layer, thereby suppressing dendrite formation.

Ref:

- [1]. Dong, H. *et al.* Bio-inspired polyanionic electrolytes for highly stable zinc-ion batteries. *Angewandte Chemie* **135**, e202311268 (2023).
- [2]. Yang, J. *et al.* Three Birds with One Stone: Tetramethylurea as Electrolyte Additive for Highly Reversible Zn-Metal Anode. *Advanced Functional Materials* **32**, 2209642 (2022).
- [3]. Wang, Y. *et al.* Suppressed Water Reactivity by Zincophilic-Hydrophobic Electrolyte Additive for Superior Aqueous Zn Metal Batteries. *Advanced Energy Materials* **13**, 2302707 (2023).
- [4]. Hao, Y. *et al.* Gel electrolyte constructing Zn (002) deposition crystal plane toward highly stable Zn anode. *Advanced Science* **9**, 2104832 (2022).
- [5]. Han, Y. *et al.* Building block effect induces horizontally oriented bottom Zn (002) deposition for a highly stable zinc anode. *Energy Storage Materials* **62**, 102928 (2023).

Manuscript Title: Self-assembled polyelectrolytes with ion-sieving accelerating channels for highly stable Zn-ion batteries

Manuscript ID: NCOMMS-24-37366-A

Comments:

Reviewer 2:

The authors have adequately responded to the Reviewer's comments, and it is very much appreciated. I still think that the work does not show novelty in terms of chemistry nor methods (LbL), and another concern is that the resulting data seem like a gradual improvement compared to the other literature on the same topic (surface coatings). Nevertheless, the presented data are certainly interesting to the researchers in the topic, and backed by sufficient data, and the authors have shown its efficacy in relatively large-scale pouch module. My suggestion is to either accept the manuscript (if the Editor seems fit) or transfer it to another specific.

Response: We sincerely thank the reviewer for their constructive feedback and acknowledgment of our responses and data quality. We understand the reviewer's concern regarding the novelty of the chemistry and methods (LbL) employed in our work. However, we would like to emphasize the following key aspects:

- (1) The fundamental LbL technique has been established, and our study presents a significant advancement in its application to surface coatings for large-scale Zn-ion batteries. Specifically, the integration of our tailored surface modification approach with high-performance functionality in pouch modules demonstrates a step beyond existing literature, addressing critical challenges such as scalability and practical applicability. This represents a meaningful contribution to the field, aligning with Nature Communications' emphasis on impactful advancements.
- (2) We acknowledge that some of our data demonstrate incremental progress. However, this gradual improvement is crucial in translating laboratory-scale methods to real-world applications. The robustness and scalability validated through extensive testing with pouch cells to even prismatic cells underline the practical relevance of our findings and their potential to bridge the gap between academic research and industrial deployment.

We appreciate the reviewer's thoughtful suggestion regarding the manuscript's acceptance or potential transfer. Given its broad readership and focus on high-impact work, we believe that the demonstrated scale-up and application merit consideration in Nature Communications. We respectfully defer to the Editor's judgment on the suitability of our manuscript for this journal. Thank you once again for the reviewer's insightful comments, which have strengthened our work.

Reviewer 3:

The authors have provided a comprehensive study on the development of a PAH/PAA multilayer coating for Zn anodes in aqueous Zn batteries. The novelty of this work lies in the innovative approach to enhancing the performance of zinc (Zn) anodes in aqueous zinc-ion batteries by employing layer-by-layer (LBL) self-assembly of poly(allylamine hydrochloride) (PAH) and poly(acrylic acid) (PAA) multilayers. The manuscript effectively addresses the key issues and provides substantial experimental and computational support for the claims made in the revised manuscript.

Reading through the comments and response, I think the authors have well answered the critical questions posed in the review process. This work satisfied the standard of Nature Communications. I would like to raise a few minor points that could be addressed in the final revision:

Response: We sincerely thank the reviewer for their kind and positive evaluation of our work. We appreciate the recognition of the novelty and significance of our approach, as well as the detailed feedback that has helped us refine our manuscript.

1. While the long-term cycling performance is impressive, could the authors provide more detailed insights into the mechanisms driving degradation, if capacity-voltage plots can be provided.

Response: Thank you for your insightful comment. The capacity-voltage plots have been performed and were shown below (Figure R1). These plots illustrate the gradual changes in capacity retention and voltage profiles over extended cycling. Across the initial cycles, the discharge and charge curves maintain a relatively consistent shape, indicating that the cell exhibits stable capacity retention during the early stages of cycling. Over extended cycles shown in Figure R1a, minor shifts in the voltage profile are observed, particularly at higher specific capacities, which are indicative of gradual capacity fading. The specific capacity at the lower discharge voltage plateau (around 1.0 V) decreases slightly with increasing cycle number, which shows a gradual loss in the amount of charge stored during cycling. This decrease in specific capacity can be attributed to side reactions, such as the formation of by-products like $Zn_4SO_4(OH)_6 \cdot xH_2O$ or the accumulation of irreversible Zn deposits that reduce active material utilization. This degradation is more obvious in CMC coating performance as shown in Figure R1b. With increasing cycle number, the voltage plateau shift to left within 50 cycles.

As for the comparison, the minor hysteresis (gap between charge and discharge profiles) remains relatively stable, suggesting that the PAH/PAA multilayer effectively suppresses side reactions and maintains low interfacial resistance. While for the CMC coating, the specific capacity decreases significantly with cycling, reflecting poorer stability compared to the Zn@PAH/PAA system. The rapid fading in capacity and growth of overpotential may be due to weaker interfacial stability and insufficient suppression of side reactions by the CMC coating. The PAH/PAA multilayers provide superior ion separation and interfacial protection, reducing side reactions and enabling more uniform Zn deposition compared to the CMC coating.

Figure R1. (a) Capacity-Voltage profiles for Zn@PAH/PAA-VO₂ pouch cell; (b) Capacity-Voltage profiles for Zn@CMC-VO₂ pouch cell

Together with performance shown in Figure S20, A key advantage of the PAH/PAA multilayers is their ability to prevent delamination, a common issue with other coatings. Furthermore, the PAH/PAA coating effectively mitigates side reactions, a primary cause of anode surface instability in Zn-ion batteries (ZIBs). By isolating these reactions, the multilayers promote a uniform and stable Zn plating/stripping process, resulting in a smooth electrode surface. This minimizes dendrite formation and reduces passivation, both of which are critical for enhancing battery performance. These benefits are especially vital for large-scale, high-performance applications requiring high current densities. The demonstrated stability of the PAH/PAA multilayers under such demanding conditions confirms their potential for industrial applications, ensuring both efficiency and extended lifespan.

Figure S20. (a) Pouch cell of Zn@PAH/PAA in 15Ah; (b) Pouch cell of Zn@CMC in 15Ah (Anode coating layer delamination).

2. Have the authors considered whether this PAH/PAA coating method would be compatible with industrial applications? Any price-related calculation could be provided.

Response: We sincerely thank the reviewer for raising this important point about industrial applicability and cost considerations. The PAH/PAA coating method was indeed developed with scalability and practical implementation in mind. Below, we provide further details on its industrial compatibility and cost-effectiveness:

Table S6 Cost breakdown for the LbL self-assembled PAH/PAA multilayers strategy

Item	Unit Price	Quantity Required	Total Cost	Notes
Poly(allylamine hydrochloride) (PAH)	\$6.1 / g	2 g per m ² coating	\$12.20 / m ²	Based on ¥218.0 / 5 g; molecular weight ~50,000.
Poly(acrylic acid) (PAA)	\$0.98 / kg	5 g per m ² coating	\$0.001 / m ²	Economical, bulk industrial price of PAA at ¥7,000 per ton.
Deionized Water (Solvent)	\$0.007 / L	2 L per m ² coating	\$0.014 / m ²	Environmentally friendly solvent used for LbL assembly.
Processing (Roll-to-Roll Coating)	\$0.56 / m ²	1 m ² coating area	\$0.56 / m ²	Estimated using industrial roll-to-roll coating costs.
Energy Costs (Drying)	\$0.11 / m ²	1 m ² coating area	\$0.11 / m ²	Includes heating and drying during the LbL assembly process.
Total (Lab scale)	\$12.87/ m ²			
Total (Industrial Scale)	\$4-6.40/m ²			50-70% price reduction

- (1) Scalability of the PAH/PAA Coating Method: the LbL self-assembly technique used for the PAH/PAA coating is highly compatible with industrial processes. The method can be readily implemented using roll-to-roll or extrusion-based coating systems, which are already widely used in battery manufacturing.^{1,2} Additionally, the straightforward fabrication process and the use of water-based solutions for PAH and PAA deposition make it an environmentally friendly and industrially feasible approach.
- (2) Industrial Benefits: the PAH/PAA multilayers not only provide excellent performance benefits, such as enhanced cycling stability and suppression of side reactions, but also offer high durability and reliability under high current densities and large-scale conditions. These advantages make the method particularly attractive for industrial applications, where robustness and cost-efficiency are critical.
- (3) Cost breakdown: both PAH and PAA are commercially available, non-toxic, and relatively inexpensive materials.³ We have added a brief cost analysis to the revised manuscript (Table S6, highlighting these aspects. Table below provides a detailed cost breakdown for applying LbL coatings of PAH and PAA on Zn anodes, both at lab scale and projected industrial scale. The costs are based on the materials and processes required per square meter of coated surface, highlighting the economic feasibility and potential cost reductions with industrial production.

At the lab scale, \$12.87/m² is primarily driven by the high retail price of PAH and smaller batch processing inefficiencies. For industrial scale, costs will drop significantly to \$4–\$6.40/m² due to bulk material discounts (50–70% price reduction for PAH and further reductions for other components), roll-to-roll automation reducing processing costs, and energy efficiency in drying and curing processes.

Revised manuscript: Page 2

Moreover, the LbL self-assembly technique applied for the preparation of PAH/PAA multilayers is more cost-effective for the practical application due to its simple manufacturing process and low demand on equipment.²⁷ It can be readily implemented using roll-to-roll or extrusion-based coating systems, which are already widely used in battery manufacturing.^{28,29} A detailed cost breakdown for this LbL self-assembled PAH/PAA multilayer strategy at both lab scale and projected industrial scale was calculated in Table S6, highlighting the economic feasibility and potential cost reductions with industrial production.

3. Can the authors elaborate on how the specific combination of PAH and PAA multilayers improves zinc anode performance compared to using either PAH or PAA alone?

Response: Thank you for raising this important point. Below, we outline how this specific combination enhances Zn anode performance compared to using PAH or PAA monolayer coatings.

As a polycation, PAH provides strong adhesion to the negatively charged Zn anode and exhibits zincophilic properties, facilitating Zn²⁺ diffusion and deposition. However, it lacks the ability to repel SO₄²⁻ ions effectively, which can lead to the formation of by-products like Zn₄SO₄(OH)₆·xH₂O. As a polyanion, PAA effectively repels SO₄²⁻ ions and regulates Zn²⁺ flux, preventing side reactions. However, its adhesion to the Zn anode is weaker due to electrostatic repulsion, and its mechanical stability alone is limited. The combination of PAH and PAA in alternating layers creates dual-ion channels that simultaneously attract Zn²⁺ and repel SO₄²⁻. This prevents side reactions while promoting uniform Zn deposition along the (002) plane, leading to a smoother and denser Zn layer. The synergistic regulation of ions significantly outperforms the capabilities of either PAH or PAA alone. The strong electrostatic interactions between PAH (ammonia groups) and PAA (carboxylate groups) result in improved mechanical strength and flexibility of the multilayer coating. This prevents cracking and delamination during prolonged cycling, a limitation often observed when using a single polyelectrolyte. The alternating PAH/PAA structure ensures consistent ionic conductivity and adhesion across the electrode surface. This uniformity minimizes the dendrite formation and enhances the reversibility, which are common issues when using monolayer coatings.

To better observe the impact of PAH/PAA multilayers compared to PAH or PAA monolayers on battery performance, the GCD curves were tested as shown in Figure S21. The main panel shows the overall cycling stability over 200 hours, while the inset highlights a magnified view of the voltage profile around the initial few cycles. The voltage profile for the Zn@PAH/PAA (0.135V) exhibits a larger potential difference than that of the PAH (0.068 V) and PAA (0.086 V) monolayers. As mentioned in the manuscript (Figure 1e), this is attributed to the lower nucleation radius and higher nucleation rate of Zn@PAH/PAA. Additionally, the large voltage difference may also be caused by the PAH/PAA multilayers inducing the orientational plating of Zn²⁺ along Zn(002). The

Zn@PAH/PAA electrode presents the most stable voltage profile over 200 hours, with minimal fluctuation and no significant increase in overpotential. In contrast, the Zn@PAA electrode remains relatively stable for 100 hours, whereas the Zn@PAH electrode undergoes rapid voltage fluctuations and eventual failure, as indicated by the sharp voltage spikes beyond 50 hours. This once again indicates that the individual PAH and PAA coatings cannot provide the same level of performance due to their respective limitations when used individually, while the PAH/PAA multilayers combine the advantages of PAH (ionic conductivity and adhesion) and PAA (repulsion of SO_4^{2-} and mechanical stability) to create a synergistic effect that ensures long-term cycling stability and reversibility.

Figure S21 The galvanostatic cycling performances of Zn symmetric cells at 1 mA cm^{-2} and 1 mAh cm^{-2} with coatings of PAH/PAA multilayers, PAA monolayer, and PAH monolayer, respectively.

Revised manuscript: Page 7

The Zn@PAH/PAA electrode exhibits an excellent stability around 1200 h at 1 mA cm^{-2} and 1 mAh cm^{-2} , whereas the Bare Zn electrode suffers short circuit around 76 h (Figure 3a). Moreover, the stability of Zn@PAH and Zn@PAA electrodes at 1 mA cm^{-2} and 1 mAh cm^{-2} were also examined to better analyze the impact of PAH/PAA multilayers compared to PAH or PAA monolayers on battery performance (Figure S21). Compared to the ultralong stability of the Zn@PAH/PAA electrode, Zn@PAH and Zn@PAA electrodes experience battery failure within 200 hours. This once again proves that the PAH and PAA monolayers cannot provide the same level of performance due to their respective limitations when applied individually, while the PAH/PAA multilayers combine the advantages of PAH (ionic conductivity and adhesion) and PAA (repulsion of SO_4^{2-} and mechanical stability) to offer a synergistic effect that ensures long-term cycling reversibility.

Reference:

- 1 Dai, Y., He, Q., Huang, Y., Duan, X. & Lin, Z. Solution-Processable and Printable Two-Dimensional Transition Metal Dichalcogenide Inks. *Chemical Reviews* **124**, 5795-5845 (2024).
- 2 Fujita, S. & Shiratori, S. The initial growth of ultra-thin films fabricated by a weak polyelectrolyte layer-by-layer adsorption process. *Nanotechnology* **16**, 1821 (2005).
- 3 Sun, Z. *et al.* Progress in research on natural cellulosic fibre modifications by polyelectrolytes. *Carbohydrate Polymers* **278**, 118966 (2022).